# Prophylactic Multi-Subunit Vaccine against *Chlamydia trachomatis:* In Vivo Evaluation in Mice

**DOI:** 10.3390/vaccines9060609

**Published:** 2021-06-06

**Authors:** Christian Lanfermann, Sebastian Wintgens, Thomas Ebensen, Martin Kohn, Robert Laudeley, Kai Schulze, Claudia Rheinheimer, Johannes H. Hegemann, Carlos Alberto Guzmán, Andreas Klos

**Affiliations:** 1Institute of Medical Microbiology and Hospital Epidemiology, Medical School Hannover, 30625 Hannover, Germany; Lanfermann.Christian@mh-hannover.de (C.L.); Kohn.Martin@mh-hannover.de (M.K.); Laudeley.Robert@mh-hannover.de (R.L.); Rheinheimer.Claudia@mh-hannover.de (C.R.); 2Institute of Functional Microbial Genomics, Heinrich-Heine-University Düsseldorf, 40225 Düsseldorf, Germany; Sebastian.Wintgens@uni-duesseldorf.de (S.W.); Johannes.Hegemann@hhu.de (J.H.H.); 3Department of Vaccinology and Applied Microbiology, Helmholtz Centre for Infection Research, 38124 Braunschweig, Germany; Thomas.Ebensen@helmholtz-hzi.de (T.E.); Kai.Schulze@helmholtz-hzi.de (K.S.); CarlosAlberto.Guzman@helmholtz-hzi.de (C.A.G.)

**Keywords:** *Chlamydia trachomatis*, vaccine, mouse model, lung infection, immune response, polymorphic membrane proteins, Pmp, Ctad1, c-di-AMP, sexually-transmitted disease

## Abstract

*Chlamydia trachomatis* is the most frequent sexually-transmitted disease-causing bacterium. Urogenital serovars of this intracellular pathogen lead to urethritis and cervicitis. Ascending infections result in pelvic inflammatory disease, salpingitis, and oophoritis. One of 200 urogenital infections leads to tubal infertility. Serovars A–C cause trachoma with visual impairment. There is an urgent need for a vaccine. We characterized a new five-component subunit vaccine in a mouse vaccination-lung challenge infection model. Four recombinant Pmp family-members and Ctad1 from *C. trachomatis* serovar E, all of which participate in adhesion and binding of chlamydial elementary bodies to host cells, were combined with the mucosal adjuvant cyclic-di-adenosine monophosphate. Intranasal application led to a high degree of cross-serovar protection against urogenital and ocular strains of *C. trachomatis*, which lasted at least five months. Critical evaluated parameters were body weight, clinical score, chlamydial load, a granulocyte marker and the cytokines IFN-γ/TNF-α in lung homogenate. Vaccine antigen-specific antibodies and a mixed Th1/Th2/Th17 T cell response with multi-functional CD4^+^ and CD8^+^ T cells correlate with protection. However, serum-transfer did not protect the recipients suggesting that circulating antibodies play only a minor role. In the long run, our new vaccine might help to prevent the feared consequences of human *C. trachomatis* infections.

## 1. Introduction

*Chlamydiae* are Gram-negative, intracellular bacteria with a unique reproductive cycle: infective elementary bodies (EBs) induce their uptake into mucosal cells where they remain in inclusions [1]. In this niche, they transform to metabolically active reticulate bodies and divide until hundreds of new infectious EBs are produced and released. In humans and animals, *Chlamydiae* cause infections of the urogenital, respiratory and gastrointestinal tract, and the eye.

### 1.1. Chlamydia trachomatis and Diseases Caused by This Intracellular Bacterium

*Chlamydia trachomatis* (C.tr.) has 19 serovars based on the major outer membrane protein (MOMP), and over 60 genotypes [2]. Serovars D-K representing the genital tract and L1-L3 representing the lymphogranuloma venereum (LGV) biovar of C.tr. lead to infections of urethra or cervix uteri. With an estimated prevalence of 4.2% in 14- to 49-year-old men and women worldwide, C.tr. is the most frequently reported bacterium causing a sexually-transmitted disease (STD). According to WHO, there were 124 million new cases in 2016 [3].

PCR studies suggest that 4.4% of sexually active 17-year-old female teenagers and 4.5% of all 18- to 19-year-old women in Germany are latently infected with C.tr. [4]. Acute genital C.tr. infections lead to few physical complaints, in particular in women. Hence, most infections remain initially unrecognized and untreated. Partially due to different serovars, C.tr. can repetitively infect the same person (reviewed in Phillips et al. [5]). It can ascend to the upper genital tract resulting in inflammation of the fallopian tubes or ovaries, or in other painful and difficult to treat pelvic inflammatory diseases. Ectopic pregnancy is one of the feared sequelae of chlamydial tissue damage [6]. More important, it has been estimated that 1 out of 200 urogenital infections leads to permanent tubal infertility [7]. Additionally, there is a relevant financial burden for society caused by often unsuccessful therapeutic attempts to later satisfy the wish to have children. Pregnant women can transmit C.tr. to newborns in the birth canal leading to conjunctivitis and pneumonia, which do not readily respond to therapy [8].

In particular men, mostly with a certain genetic background, can suffer from reactive arthritis [9]. In this rheumatic disease, persistent viable C.tr. with reduced metabolism are detectable in the synovia of the affected joints for up to several years. Additionally, there is an ongoing discussion about a potential link between C.tr. infections and urogenital tumors [10].

In developing countries, this STD is even more frequent. Under these circumstances, routine PCR screenings of young women in order to manage the disease and to avoid its sequelae is impracticable. Even in industrialized countries, only screening for C.tr.-DNA during pregnancy has been successful [11]. Attempts to change human risk behavior and to decrease infection rates by information campaigns and by providing for all woman ≤25 years the opportunity for an annual PCR screening paid by the health insurance remained ineffective.

The serovars A–C forming the trachoma biovar of C.tr. are responsible for this neglected tropical disease [12,13]: According to the WHO Global Health Observatory 2019, trachoma remains a public health problem in 43 countries [14]. It is responsible for partial or total visual impairment of 1.8 million people. In 2017, 83.5 million people received antibiotics for trachoma, and 231,000 people received surgery for its late blinding stage, trachomatous trichiasis. Originally, the WHO aimed to eliminate trachoma by 2020. Apparently, with the so far applied SAFE Strategy (Surgery, Antibiotics, Facial cleanliness, Environmental improvement) alone, it will be impossible to achieve this ambitious, but highly unrealistic aim, even in the coming years.

Obviously, there is an urgent need to develop a vaccine against C.tr. that prevents urogenital infection with its sequelae, in particular female infertility, and, additionally, one preventing trachoma. However, there is still no such vaccine available.

### 1.2. Former Attempts to Develop a Vaccine against Chlamydia trachomatis

According to the review “Seventy Years of Chlamydia Vaccine Research-Limitations of the Past and Directions for the Future” by Phillips et al., 78 studies have been performed with C.tr. between 1949 and 2017 [5]. Its main conclusion is that “… no single antigen type or target, adjuvant, or route of administration has been established as a clear front-runner for effective vaccination.” [5]. Unfortunately, most attempts of vaccine development have been unsuccessful, either due to missing (cross-serovar) protection or even due to increased immunopathology during a subsequent chlamydial infection (reviewed in: [15,16,17,18,19]). An extensive presentation or discussion of all these vaccine candidates goes beyond the scope of the manuscript, thus, we focused our analysis on the most successful or, in regard to this project, most relevant vaccination studies.

In several clinical trachoma studies with live or inactivated EBs of C.tr., transient reduction of the disease and a reduced bacterial load occurred. However, discouraging studies led to the conclusion that this type of vaccination “might lead to a more severe disease upon challenge with a heterologous serovar”, and that the protection achieved is only short-lived and possibly serovar or serogroup specific (as reviewed in [19]). Presently, it is under investigation whether or not attenuated C.tr. isolates, such as the plasmid-deficient derivative of a C.tr. D strain, which is protective against genital tract infections of Rhesus macaques [20], can be used in humans (reviewed in [19]). However, there are safety concerns and challenges attached to the manufacturing of a whole cell Chlamydia vaccine according to industrial good manufacturing practice standards [20,21].

Thus, many researchers have shifted their efforts to potential C.tr. subunit vaccines. As reviewed in [19], the majority of them are directed against surface exposed antigens such as purified native nMOMP or recombinant rMOMP or polymorphic membrane proteins (rPmp’s), but others also targeted intracellular antigens, such as plasmid-encoded glycoprotein 3 or chlamydial protease-like activity factor.

Several investigations concern the nine Pmp’s A to I that form the largest protein family in C.tr. [22]—with four of them being part of the multi-subunit vaccine used in the present study. These proteins of ~100 to 150 kDa share autotransporter characteristics with a short N-terminal Sec signal sequence (24 to 50 amino acids, aa), a large passenger domain (PD, ~650 to 1200 aa), and a C-terminal β-barrel (~300 aa) for outer membrane translocation. Evidence suggests that Pmp proteins form functional homomeric oligomers [23]. Pmp’s from C.tr. act as adhesins and are essential for infection [24]. Indeed, a C.tr. mutant strain harboring a *pmpD* null mutant shows reduced attachment to epithelial cells in vitro and in a non-human primate macaque model of ocular infection a 13-fold reduced chlamydial burden in comparison to the wild-type strain [25]. Pmp21, the *C. pneumoniae* homolog of the C.tr. PmpD, has been shown to be an adhesin and invasin, which is binding to and activating the epidermal growth factor (EGF) receptor as part of the EB internalization process [26].

Antibodies to PmpD neutralize all C.tr. serovars in vitro, and PmpD as antigen leads to protection in various animal systems [27]. Vaccination of mice with short PD protein fragments from PmpE, F, G, and H from *C. muridarum* combined with DDA/MPL as adjuvant confers protection against *C. muridarum* after vaginal challenge [28]. Moreover, PmpE, F, G, and H fragments plus MOMP from C.tr. D protect against a C.tr. challenge infection in mice [29]. Based on the high protein identity between C.tr. serovars and *C. muridarum*, short protein fragments from the PD from all nine C.tr. serovar E Pmp’s were tested in mice using subcutaneous (s.c.) application and CpG-1826 and Montanide ISA 720 VG as adjuvants. Indeed, this results in limited cross-species protection against a subsequent *C. muridarum* infection [30]. A PD fragment from PmpC from C.tr. B applied by s.c. and intramuscular (i.m.) route also cross-protects against a challenge with *C. caviae* in the guinea pig [31]. Similarly, partial cross-protection was observed for a PmpA PD fragment from C.tr. E plus CpG-ODN 1826 adjuvant against a subsequent *C. muridarum* infection. Yet, here, adjuvant-induced immunopathology upon challenge of the genital tract was reported [32]. Thus, Pmp proteins can protect against a C.tr. infection and show even partial cross-species protection. However, limited success or immunopathology has been often an issue.

### 1.3. Selection of Chlamydial Antigens, the Adjuvant and the Test System for Our New Vaccine

We selected for our new five-component subunit vaccine (5cVAC) five promising candidate antigens from C.tr. serovar E in their recombinant, purified form (see Table 1). Four antigens are members of the Pmp adhesin protein family. The fifth antigen is Ctad1, a novel adhesin which exploits as invasin the human integrin β1 receptor [33]. So far, it has not been tested as a vaccine antigen.

The plethora of former vaccination studies showed also that not only the antigens but also adjuvant and route of application are critical for the success, as recently reviewed [34,35]. Although still a controversial issue, most experts speculate that a protective vaccine against Chlamydia might require a combined humoral and cellular immune response [19].

Cyclic-di-adenosine monophosphate (c-di-AMP) is a second messenger in prokaryotes that exhibits strong immune modulatory properties. It was selected as adjuvant because it can stimulate antigen-specific antibody and Th1/Th2/Th17 T cell responses, as well as CD8 cytotoxic T lymphocytes by promotion of cross-presentation. Moreover, c-di-AMP is suitable for mucosal and systemic application, and, when administered by the mucosal route, leads to mucosal responses at local and distant mucosal territories. The c-di-AMP also promotes a self-limited immune activation, which is restricted to the administration site, resulting in an enhanced safety profile [36,37,38,39].

Finally, it is not clear which surrogate marker of the immune response could actually best replace observable protection. Selecting only immune-dominant components for the vaccine without verification of the protective potential carries the risk to choose a non-protective or to reject erroneously an essential one. Hence, instead of using immunological surrogate markers only suggesting protection, we determined efficacy directly, i.e., the protection achieved by the adjuvanted vaccine in our relatively fast-to-perform and highly quantifiable mouse vaccination-lung challenge C.tr. infection model [40]. We have already used this model for successful identification of a member of a new class of C.tr.-directed antibiotics [41]. With its short challenge phase of only 7 days, we have a suitable screening method at our disposal, where one can easily quantify the main defining parameters of protection.

The purpose of this study was to identify by broad analysis a useful combination of adjuvant and antigens, which successfully targets several serovars of the human pathogen C.tr. in lung infection. We wanted to evaluate the protection achieved after vaccination by two different application routes, and to get insights in the induced humoral and cellular immune responses in the lung infection model. Thereby, we wanted to lay the foundation for confirmatory future studies challenging vaccinated mice with C.tr. by the transcervical-intrauterine route and determining achieved protection in regard to chlamydia-induced tissue-reorganization or infertility.

Indeed, after mucosal application, our new multi-subunit vaccine led in the lung infection model to a high degree of protection against C.tr. E, and cross-serovar protection against serovar D from the urogenital, L2 from the lymphogranuloma and serovar A from the trachoma biovar. Moreover, we found long-lasting 5cVAC-specific antibodies and, most likely, in our case functionally more important, T cell responses in the vaccinated animals.

## 2. Materials and Methods

### 2.1. Chlamydial Strains and Their Growth in Cell Culture

For the mouse vaccination-lung challenge infection experiments, the following strains of different C.tr. serovars were propagated in baby hamster kidney 21 cells (BHK-21, ATCC: CCL-10) as previously described [40]: the genital serovars E, strain DK20 (Origin: Institute of Ophthalmology, London), D, strain UW3/CX (ATCC: VR-885), and L2, strain LGV II 434 (ATCC: VR-902B), as well as the ocular C.tr. serovar A, strain HAR-13 (ATCC: VR-571B; a kind gift of S. Birkelund, Aarhus, Denmark).

Inclusion-forming units (IFU) of the stock-solutions were assessed by repetitive titration using cervix epithelial HeLa-T cells, kindly provided by R. Heilbronn, Berlin, Germany [42]. For negative controls, i.e., ‘mock’ infection, BHK-21 cells were identically processed as our chlamydial stock preparations, however, without the addition of any bacteria. All chlamydial and the mock preparations were Mycoplasma-free tested by PCR.

### 2.2. Preparation and Purification of the Recombinant Antigens PmpA, PmpD, PmpG, PmpH, and Ctad1 Derived from C.tr. Serovar E/DK20

In most experiments, combined with an adjuvant (see below), an equimolar mix of the following five purified, recombinant antigens derived from serovar E of C.tr. was applied (5cVAC): PmpA, PmpD, PmpG, PmpH, and Ctad1.

Our 5cVAC antigen composition consists of PD protein fragments from PmpA, D, G, and H. We succeeded in producing full-length PD fragments from PmpA (protein fragment aa 53 to 665), PmpG (aa 29 to 637), and PmpH (aa 26 to 690), while for PmpD only the first half of the large PD (aa 34 to 619) could be generated in sufficient amounts. Full-length Ctad1 (aa 1 to aa 433) is the fifth component of 5cVAC. All recombinant antigens are derived from serovar E of C.tr. One experiment was performed with 2cVAC, i.e., an equimolar mix of the two antigens PmpD and Ctad1, only. See Table 2 for protein IDs and lengths of the complete proteins (aa) of the five antigens of 5cVAC from *C. trachomatis* E/DK20, and their homologous proteins in C.tr. D, A, L2, and *C. muridarum*.

Gene fragments encoding PDs of the four Pmps were PCR-amplified from genomic DNA of C.tr. serovar E strain DK20 and were cloned into the single-cut expression vector pKM32 via homologous in vivo recombination in yeast strain CEN.PK 2-1C [43]. Subsequently, the resulting plasmids were amplified in *E. coli* strain XL1 blue and verified via control restriction enzyme digests followed by gel electrophoresis and sequencing. Next, the four *pmp* containing plasmids as well as pST42 (carrying *ctad1*; [33]) were transformed into our expression strains *E. coli* BL21 (Pmp’s) and *E. coli* Origami (Ctad1).

Protein expression was induced in liquid culture (usually 1000 mL) using 1 mM IPTG for 4 h at 37 °C. Cells were collected by centrifugation (5000 rpm). Pelleted cells were lysed under denaturing conditions using a lysis buffer (6 M guanidine-HCl/20 mM Tris HCl/0.5 M NaCl/1 mM β-mercaptoethanol) overnight at 4 °C. On the following day, the lysate was sonicated on ice for 1 min. Insoluble debris was removed by centrifugation at 24,000 rpm for 1 h. Recombinant proteins were subsequently purified by affinity chromatography under denaturing conditions.

**Table 2 vaccines-09-00609-t002:** Protein ID (NCBI) and lengths of complete proteins in amino acids (aa) of the five antigens of 5cVAC which are derived from *C. trachomatis* E, and their homologous proteins in C.tr. A, D, L2, and *C. muridarum*. The strain names of the different serovars from which this information is derived are included in the column titles. Source NCBI: National Center for Biotechnology Information (NCBI) [44].

Pmp’s	C.tr. E E/DK20	C.tr. *D* *UW3/CX*	C.tr. A A/HAR-13	C.tr. LGV L2/434/Bu	*C. muridarum* Nigg 2 MCR
**A**	WP_009872639.1 (975aa)	WP_010725190.1 (975aa)	WP_009871764.1 (975aa)	WP_009872639.1 (975aa)	WP_010231239.1 (976aa)
**D**	WP_009872948.1 (1530aa)	WP_010725352.1 (1531aa)	WP_011324878.1 (1531aa)	WP_009873422.1 (1530aa)	WP_010229791.1 (1520aa)
**G**	WP_014541208.1 (1013aa)	WP_010725377.1 (1013aa)	WP_011324914.1 (1013aa)	WP_009873478.1 (1012aa)	WP_010229977.1 (986aa)
**H**	WP_014541209.1 (1014aa)	WP_010725378.1 (1016aa)	WP_011324915.1 (1018aa)	WP_009873479.1 (1006aa)	WP_010229979.1 (980aa)
**Ctad1**	WP_010724985.1 (433aa)	WP_010724985.1 (433aa)	WP_011324521.1 (433aa)	AGJ64368.2 (477aa)	AID37821.1 (446aa)

Soluble fractions containing N-terminal (Pmp’s) or C-terminal (Ctad1) His-tagged proteins were loaded onto HiTrap chelating HP columns (GE Healthcare). Bound His-tagged proteins were sequentially washed with buffer B (8 M urea/0.1 M NaH_2_PO_4_/10 mM Tris HCl/1 mM β-mercaptoethanol (not for Ctad1)/20 mM imidazole; pH 8), and buffer C (8 M urea/0.1 M NaH_2_PO_4_/10 mM Tris/HCl/1 mM β-mercaptoethanol (not for Ctad1)/40 mM imidazole; pH 6.3). Proteins were eluted with 8 M urea/0.1 M NaH_2_PO_4_/10 mM Tris/HCl/1 mM β-mercaptoethanol (not for Ctad1)/500 mM imidazole (pH 6.3). For renaturation, eluted proteins (PmpG, PmpH, PmpD, Ctad1) were dialyzed three times against PBS (pH 7.4) at 4 °C over two nights. As this procedure caused PmpA to precipitate, PmpA was renatured using Amicon centrifugation. In this process, 1 mL of elution fraction was mixed with 10 mL of PBS + 200 mM arginine, loaded onto an Amicon^®^ Ultra-15 Centrifugal Filter unit and centrifuged at 4 °C as recommended by the manufacturer, until the volume was reduced to 1 mL. This step was repeated once. At last, endotoxin was reduced by the use of endotoxin removal columns (ToxinEraser Endotoxin Removal Kit). Coomassie-stained SDS-Page and Western blot of the purified recombinant antigens used in the vaccine formulation are depicted in Appendix A. The concentration of each recombinant protein was determined via Bradford assay and adjusted to a minimum of 700–1000 µg/mL.

### 2.3. Immunoblotting and Coomassie-Staining

SDS-PAGE and immunoblotting were performed as described previously [45]. Recombinant proteins were detected with monoclonal anti-His (Qiagen, Hilden, Germany) antibodies and visualized with AP-conjugated antibody (Sigma, Munich, Germany). Gels were stained with Coomassie Brilliant Blue G250 (Serva, Heidelberg, Germany) (Appendix A).

### 2.4. Preparation of the c-di-AMP-Adjuvanted Vaccine Directly before Application to Mice

In all mouse vaccination experiments (i.e., with 5cVAC and 2cVAC), a final volume of 30 µL containing 3.95 × 10^−11^ mol of each LPS-reduced/free antigen (summing up to approximately 12.5 µg protein for 5cVAC in total) and 10 µg of the adjuvant c-di-AMP was administered to the mice. For the experiment with 2cVAC, 3.95 × 10^−11^ mol of PmpD and Ctad1 were administered with the identical amount of c-di-AMP in the identical total volume. Under sterile conditions and on ice, the vaccine components were mixed and vortexed thoroughly. For dilution PBS/L-arginin 200 mM was used. Aliquots of the prepared stock solutions of 5cVAC and 2cVAC were stored at −80 °C.

According to the total amount needed on the day of vaccination, several aliquots of the antigen mix were thawed and pooled. Directly before application the adjuvant c-di-AMP in H_2_O was added 1 in 30. This solution was kept on ice until application. Corresponding to that, as negative controls, 1 µL c-di-AMP (without antigens) in 30 µL buffer, or 30 µL buffer alone, respectively, were administered per mouse.

### 2.5. Three Experimental Setups of the Mouse Vaccination-Chlamydia Trachomatis Lung Challenge Infection Model: Short-Term, Serum-Transfer, and Long-Term Protection

The phase of vaccination with adjuvanted 5cVAC (or 2cVAC) was identical in all three experimental setups (Figure 1a–c). The hormonal cycle of seven-week-old female C57BL/6J mice was synchronized by intraperitoneal (i.p.) application of 2.5 mg medroxyprogesterone acetate in 200 µL 0.9% NaCl, in order to diminish hormone dependent variations, and additionally to facilitate comparison with results of a potential urogenital infection model in the future. To induce protection, the mice were vaccinated three times with 30 µL of c-di-AMP-adjuvanted 5cVAC, or the adjuvant c-di-AMP alone, or buffer, respectively, at day 7, 21, and 28. Vaccination was usually performed by the intranasal (i.n.) route. This application route was also selected since it has been demonstrated that it triggers similar local immune responses in both, the respiratory and the urogenital tracts [46,47].

In a few experiments, the adjuvanted 5cVAC was applied not by i.n. but alternatively parenterally by s.c. route (as indicated in the corresponding figure).

(a)Short-term protection model (Figure 1a): in week 7 of the experiment, the cycle of the female mice was hormone-synchronized again. In week 8, i.e., 4 weeks after the last booster vaccination, the 15-week-old mice were challenged by i.n. application of Chlamydia differently pretreated animals per group for the C.tr. E, D, and L2, and mice per group for C.tr. A, respectively). One group of animals received mock material instead of Chlamydia (usually *n* = 8, only in one experiment (as indicated) *n* = 4).

Preliminary i.n. titration-experiments had been performed to identify the optimal amount of the different C.tr. serovars for the main experiments (Appendix A). In the lung-pneumonia model this can be done by daily observation of body weight, clinical score, and survival, i.e., time until the humane endpoint (see Appendix A) is reached, and the mice have to be sacrificed painlessly. In order to observe vaccine-induced protection with high sensitivity, C.tr. infected mice had to exhibit a severe, but still tolerable course of infection. Chlamydial infections lead to a protracted course of disease, and therefore, they usually recover even when the body weight is reduced by >20%. Hence, our permit from the corresponding authorities allows even then (under close supervision) continuation of observation. Yet, we had to avoid transient weight loss of >25–30%, and the survival rate should remain in the range of 90%. In any case, most animals had to reach the end of the observation period for the intended analysis of their blood and lung tissue. When we originally established this mouse model for C.tr. [40], we had seen that a time frame of 7 days is optimal. During the following days, most infected and severely sick C57BL/6J mice recover little by little (see also Appendix A) until infectious Chlamydia are no longer detectable in lung homogenate after 2–3 weeks [40]. Notably, if longer observation periods were chosen, some mice would stay ill and would prematurely reach the humane endpoint due to exhaustion. To induce a similar severity of resulting pneumonia, according to the pre-tests (Appendix A), different amounts of inclusion forming units (IFU) of the genital serovars E, D, L2 and the ocular serovar A of C.tr. per mouse had to be used: 1.3 × 10^6^ for serovar E, 2 × 10^6^ for D, 4 × 10^5^ for L2, and 4 × 10^6^ for A.

(b)Serum transfer-short-term protection model (Figure 1b): We wanted to clarify to which extent antibodies that are circulating in murine blood after vaccination with c-di-AMP-adjuvanted 5cVAC are actually conferring protection against C.tr. E. Four weeks after the last i.n. or s.c. application of the vaccine, or buffer, or c-di-AMP alone, respectively, i.e., exactly at the time point when the challenge infection was normally performed, the mice were sacrificed and their blood was collected by heart puncture. The sera of the four differently treated groups of animals (*n* = 10–14 per group) were pooled and 170 µL were transferred twice intravenously (i.v.) to non-vaccinated 14- to 15-week-old healthy animals (*n* = 5–7 per group) on days 4 and 5 after their hormone-synchronization. C.tr.-specific antibodies were determined in the four donor pool sera as well as in the sera of the individual recipient animals. For that purpose, small blood samples were collected on the day between serum-transfer and i.n. C.tr. E-challenge infection. One week after hormone-treatment and two days after the last serum-transfer, i.n. challenge infection was conducted with 1.3 × 10^6^ IFU of C.tr. E, i.e., exactly like in the short-term protection model.(c)Long-term protection model (Figure 1c): To determine the long-term protection achieved, mice that had been vaccinated i.n. with c-di-AMP-adjuvanted 5cVAC in week 1, 3, and 4, as described above, were hormone-synchronized again in week 23. Twenty weeks after the last booster vaccination, the mice were challenged i.n. with C.tr. E. According to preliminary titration experiments (Appendix A), 31-week-old mice react more sensitive to chlamydial infection, independent of body weight or vaccination. Thus, a more than three-times lower IFU (4 × 10^5^) of C.tr. E had to be administered for challenge in this setting. Otherwise, a relevant portion of the older non-vaccinated, infected mice would not have achieved the desired seven-day-observation period after challenge infection, thereby reducing the group size of this essential control group.

The exact number of mice per group for each experiment is depicted in Appendix A, and summed up as a range in each legend. For both, i.n. application of the vaccine and i.n. C.tr. challenge, mice were anesthetized by i.p. injection of 0.1 mL anesthetic solution (100 mg/kg BW Anesketin, 4 mg/kg Rompun in 0.9% NaCl) per 10 g body weight.

The observation period after chlamydial challenge infection was seven days in all three related models. In order to reduce variation within experiments with a specific serovar as much as possible, mice of the vaccinated and the non-vaccinated control groups were housed in mixed cages and handled in parallel. The depicted results usually combine data obtained in two or three (and four for the long-term experiment) staggered, otherwise identical experiments. Mice were monitored closely and assessed daily using humane endpoint defining criteria. On the last day of challenge infection and developing pneumonia, all mice were painlessly sacrificed for further analysis of their lung (and partially also their spleen) as well as their blood which was drawn by heart-puncture.

### 2.6. Clinical Scoring

The clinical score was recorded daily after challenge infection. It is based on pilo-erection, body posture, locomotion, breathing, dehydration, attention/curiosity and secretion from nose and eyes (for details see Appendix A).

### 2.7. Determination of Bacterial Load in Mouse Lung Homogenate by Flow Cytometry

Homogenates from the right lung lobes were harvested as described elsewhere [48]. The determination of the different C.tr. strains in the cryopreserved homogenate was performed by flow cytometry as previously described by us for *C. psittaci* [49]. In brief, cryopreserved lung homogenates were thawed 15 min on ice. The samples were vortexed for 3 min and centrifuged for 15 min, 500× *g*, 4 °C. HeLA–Cells were infected with the serial diluted supernatant and cultured for 16–20 h. For staining of intracellular Chlamydia and the cells, Pathfinder^®^ (Fluorescein-conjugated murine monoclonal antibody to chlamydial LPS; 0.1% Evans Blue; Bio-Rad, Hercules, CA, USA) in PBA-S (PBS supplemented with 0.25% BSA, 0.5% saponin for permeabilization) was used. To calculate bacterial content (IFU) in the samples with lung homogenate, as internal standard, a dilution series of the corresponding Chlamydia stock preparation with known concentration was analyzed in parallel.

### 2.8. Determination of TNF-α and IFN-γ and Myeloperoxidase Levels in Lung Homogenate

Concentration of the key cytokines TNF-α and IFN-γ and the granulocyte marker myeloperoxidase (MPO) were determined in the mouse lung homogenate by ELISA according to the manufacturers protocol (TNF-α: ELISA MAX™ Deluxe Set Mouse TNF-α, BioLegend, 430904; IFN-γ: ELISA MAX™ Deluxe Set Mouse IFN-γ, BioLegend, 430804; MPO: MPO, Mouse, ELISA kit, Hycult Biotech, HK210-02). Absorbance was measured at 450 nm/540 nm (Synergy HTC Multi-Mode Reader Biotec^®^ plate reader).

### 2.9. ELISAs for Circulating 5cVAC-Specific IgA/M/G and IgG Subtypes in Blood

The levels of 5cVAC-specific Igs were determined by ELISA (modified from a recently described Ig-ELISA using homogenate of *C. psittaci* as antigen [49]) in mouse EDTA-plasma or serum, as indicated in the graphs. Polystyrene microtiter plates were pre-coated at 4 °C for 16 h with 100 µL of 1 μg/mL equimolar 5cVAC antigen mix per well in PBS, or the equimolar amount of only one of the recombinant antigens, as indicated in the graphs, respectively. Non-specific binding-sites were blocked by PBS containing 1% BSA and 5% sucrose.

After washing the pre-coated plate, a serial dilution of high-titer standard plasma (see below) or pre-diluted plasma or serum samples were added for 1.5 h at 37 °C. The specimens were collected before and/or 7 days after challenge infection, as indicated. Only a small amount of blood can be drawn (according to local authorities and GV-SOLAS) from animals a few days before infection. Thus, based on a pilot test (data not shown), all respective EDTA-plasma samples were diluted 1:316 and higher for determination of antigen-specific IgA and IgM, or 1:3160 and higher for quantification of antigen-specific IgG, respectively. For the analysis of blood obtained by heart puncture from sacrificed mice at the end of the seven-day observation period, less limiting serum or even pool serum could be analyzed at a lower dilution.

After washing, the following biotinylated secondary antibodies were applied: IgA-rat anti-mouse (α-chain), HRP, AK-Online, ABIN135043, 1:5000; IgM-rat-anti-mouse, HRP, BD Biosciences, 550588, 1:1000; total IgG-F(ab’)2 goat anti-mouse IgG, HRPO, Dianova, 115-036-062, 1:10,000; IgG_1_-rat-anti-mouse, HRP BD Biosciences, 559626, 1:1000; IgG_2a_-rat-anti-mouse, HRP, BD Biosciences, 553391, 1:1000; IgG_2b_-goat anti-mouse, HRP, Invitrogen M32407, 1:2000. Next, 10 μg/mL streptavidin conjugated HRP (Jackson Immuno Research, 016-030-084) was used. Plates were washed again and incubated with substrate buffer (90 mM Na-acetate, 90 mM citric acid, 100 μg/mL TMB, 0.0045% H_2_O_2_) at RT in the dark. After 20 min, the enzymatic reaction was stopped and photometric absorbance was determined at 450 nm/540 nm (Synergy HTC Multi-Mode Reader Biotec^®^ plate reader).

Arbitrary units (U) were calculated for each Ig-class in relation to a standard, serial diluted high-titer pool plasma. The standard plasma (stored in frozen aliquots) had been collected from mice after vaccination with adjuvanted 5cVAC and subsequent challenge with C.tr. E. Arbitrary units were determined by the inverse function of the four parametric logistic equation, using iterative curve fitting of the standard curve (GraphPad Prism, GraphPad Software, San Diego, CA, USA).

### 2.10. Functional Analysis of Mouse Splenocytes: Proliferation, ELISpot, Flow Cytometry

For the determination of cellular proliferation, ELISpot assay and multifunctional flow cytometric analysis of T cells, the spleens of mice (pretreated with adjuvanted 5cVAC, or adjuvants, or buffer, respectively) were aseptically removed. For further analysis, cell suspensions of spleens of each mouse were freshly prepared and erythrocytes were lysed.

#### 2.10.1. Measurement of Cellular Proliferation

The splenocytes were cultured in the presence of different concentrations of 5cVAC antigen (0.5–10 µg/mL) or buffer as negative control, as indicated. Positive controls received 5 µg/mL concanavalin A. The ability of immune cells derived from spleen to proliferate upon restimulation with 5cVAC as well as their cytokine profile were measured 96 h post restimulation. The proliferative activity was determined by the incorporation of [^3^H]-thymidine (CPM). Beta-emission was measured using a scintillation counter (Wallac 1450, Micro-Trilux).

#### 2.10.2. ELISpot Assay

The number of antigen-specific cytokine-producing splenocytes was determined using an ELISpot assay as previously described [50]. 96-well plates (BD Pharmingen, San Diego, California) were coated with anti-IFN-γ, anti-IL-2, anti-IL-4, or anti-IL-17 antibodies overnight at 4 °C. Then, plates were washed once with culture medium (RPMI, 10% FCS, PenStrep, L-glutamine, and β-mercaptoethanol), and cells were seeded in culture medium with or without 5cVAC (1 µg/mL). Plates were incubated 24 h for IFN-γ and 48 h for the other cytokines. Then, cells were removed and the plates processed according to manufacturer’s instructions. Colored spots were counted with an ELISpot reader (CTL-Europe GmbH) and analyzed using the ImmunoSpot image analyzer software v3.2. In order to determine the number of cytokine-secreting cells in the spleen, murine IFN-γ, IL-2, IL-4, and IL-17 ELISpot kits (BD Pharmingen) were used according to the manufacturer’s instructions. Colored spots were counted with an ELISpot reader (C.T.L.) and analyzed using the ImmunoSpot image analyzer software v3.2.

#### 2.10.3. Flow Cytometric Determination of Multifunctional T Cells

Splenocytes (2 × 10^7^ cells per mL) were incubated (37 °C, 5% CO_2_) in RPMI containing the 5cVAC antigen or buffer to determine the basal cytokine production. Viable singlet leukocytes were gated for CD3^+^, CD4^+^, CD8^+^ and subsequently analyzed for the expression of intracellular IL-2, IL-4, IL-10, IL-17, TNF-α, and IFN-γ as previously described in Landi et al. [37].

### 2.11. Group Sizes and Statistical Analysis

In the majority of cases, logarithmic transformation of parametric data was performed in order to accomplish Gaussian distribution. For statistical analysis, the following tests have been used: two-way ANOVA with Bonferroni post-test for comparison of body weight and Kruskal–Wallis test with Dunn’s multiple comparison post-test (≥3 groups) or Mann–Whitney *t*-test (2 groups) for the clinical score. For bacterial load, MPO, IFN-γ, and TNF-α, one-way ANOVA with Bonferroni’s multiple comparison were used. For cellular proliferation, ELISpot and multifunctional T cell analysis, the statistical significance of the differences observed between the different experimental groups was analyzed using two-way ANOVA followed by Tukey’s multiple comparisons test. Statistics were calculated using the GraphPad Prism software version 5.02 (and 8.4.0 for data obtained on splenocytes) for Windows, GraphPad Software, San Diego, CA, USA. Differences were considered significant at *p* < 0.05 or lower. If not indicated otherwise, *–**** indicate statistical significances between the 5cVAC group and the buffer control group with *p* < 0.05, <0.01, and <0.001, *p* < 0.0001, respectively.

## 3. Results

### 3.1. Intranasal Vaccination with the c-di-AMP-Adjuvanted 5cVAC-Formulation Improves Loss of Body Weight and Clinical Score in Lung Challenge Infection with Serovar E, D, L2, or A of C. Trachomatis

Female eight-week-old C57BL/6J mice received c-di-AMP-adjuvanted 5cVAC three times within three weeks via the i.n. route (Figure 1a: short-term protection model). As negative controls, animals were pretreated in parallel with the adjuvant alone, or buffer. Four weeks after the last booster vaccination, the 15-week-old animals were infected i.n. with 1.3 × 10^6^ IFU of C.tr. E, or with serovars D, L2 and A, respectively. To induce with these serovars a similar severity of the resulting lung disease, different amounts (IFU) in the range from 4 × 10^5^ to 4 × 10^6^ IFU had to be used (see also Section 2). Some control mice received Chlamydia-free mock material instead.

During 7 days of C.tr. E challenge infection, body weight (%) (Figure 2a1) and clinical score (Figure 2a2) were assessed daily. Intriguingly, with a delay of 2–3 days, there was a drastic improvement in body weight and clinical score in i.n. vaccinated mice demonstrating strong vaccine-induced protection. Moreover, according to this course of the disease-characterizing parameters, the vaccinated animals exhibited similar cross-serovar protection against the genital C.tr. serovars D and L2, as well as the ocular serovar A (Figure 2b1–d2).

### 3.2. Intranasal Vaccination with c-di-AMP-Adjuvanted 5cVAC Improves Bacterial Clearance and Leads to Decreased Levels of the Granulocyte Marker MPO in the Lung after Challenge Infection with four Different Serovars of C. trachomatis

On day 7 after challenge infection with the four different C.tr. serovars E, D, L2, or A, respectively, the i.n. vaccinated mice as well as the animals of the two control groups were sacrificed to determine the amount of viable, infectious Chlamydia (IFU) and the level of the granulocyte marker MPO in their lung homogenate (Figure 3). After challenge with C.tr. E, the chlamydial load in the lung of i.n. vaccinated mice was approximately 100-fold lower as compared to both non-vaccinated control groups. Remarkably, a drastic reduction of the bacterial load could be seen after challenge infection with serovar D and A. The difference of vaccinated to (buffer) control mice was still significant after challenge with serovar L2, but less striking (Figure 3a1–d1). The level of the granulocyte marker MPO in lung homogenate was also lower in vaccinated mice after challenge infection with C.tr. E, D, and A (Figure 3a2,b2,d2), suggesting diminished granulocyte driven inflammation. The improved clearance of Chlamydia from the lung and the lower level of the granulocyte marker after i.n. application indicate not only that this form of repetitive mucosal application of c-di-AMP-adjuvanted 5cVAC protects against C.tr. E, but also that this vaccine induces cross protection against three additional, different serovars.

### 3.3. Intranasal Vaccination with Adjuvanted 5cVAC Leads to Decreased Levels of the Key Cytokines TNF-α and IFN-γ in Lung Homogenate after Challenge Infection with Different Serovars of C. trachomatis

As part of the innate immune response against the intracellular pathogen, the cytokines TNF-α and IFN-γ are elevated in Chlamydia-infected lungs [40]. Thus, on day 7 of challenge infection with the four different C.tr. serovars, the levels of these two cytokines were determined (Figure 4a1–d1,a2–d2) in the lung homogenate of vaccinated and non-vaccinated control mice. Depending on i.n. vaccination with adjuvanted 5cVAC, lower levels of the two key cytokines were present after challenge infection.

### 3.4. After Challenge Infection with Serovar E of C. trachomatis, s.c. Vaccination with c-di-AMP-Adjuvanted 5cVAC Does Not Lead to Relevant Protection in Regard to Body Weight, Clinical Score, Bacterial Load, the Levels of the Granulocyte Marker MPO, TNF-α or IFN-γ in Lung Homogenate

In contrast to the positive findings obtained after i.n. application of c-di-AMP-adjuvanted 5cVAC, s.c. vaccination had, even after challenge with the serovar E C.tr. strain that is closest related to the vaccine, no effect on the elevated clinical score, and only on day 7 a minimal ameliorating effect on weight loss (Figure 5a,b). Moreover, s.c. vaccination did not have any positive effect on the amount of viable C. tr. E or the elevated level of the granulocyte marker MPO in the lung (Figure 5c,d). In addition, we could not detect decreased levels of TNF-α or IFN-γ caused by s.c. vaccination (Figure 5e,f). These findings prove the importance of the i.n. route of 5cVAC application.

### 3.5. IgA, IgM, and Total IgG antibody Response Profiles towards the Adjuvanted 5cVAC Antigen-Mix or the Single Vaccine Components in Blood of Five Mice after Vaccination with 5cVAC, Seven Days before or after C.tr. E Challenge, Respectively

To get an impression of individual humoral immune responses against the five different antigens, plasma of the first five mice which were vaccinated i.n. with 5cVAC was analyzed 7 days before and 7 days after C.tr. E challenge infection (Figure 6). Due to limited blood volume which can be drawn from mice during an ongoing experiment and the broad analysis performed with each sample (18 ELISAs for three types of Igs with six different antigens), the plasma had to be pre-diluted to 1:316 for antigen-specific IgA and IgM, or 1:3160 for IgG, respectively. Elevated levels [U] of IgA or IgM correspond in this experimental setup to titers of 1:300 to 3000, and elevated units of IgG to titers of 1:3000 to 30,000, or higher (data not shown, see also Figure 7). All negative controls with plasma of individual non-vaccinated C.tr. E-challenged or non-challenged mice remained below <1 U (data not shown).

To overcome the limiting amount of plasma of individual animals and to be able to detect also potentially lower antigen-specific IgG levels (at least in the larger amount of blood obtained by heart puncture seven days after infection) the sera of *n* = 9 vaccinated, challenged and euthanized mice were pooled. This larger volume of pool serum could be analyzed with less diluted samples, i.e., starting a dilution curve (now also) for IgG already at 1:316 (Figure 7).

The specific humoral responses against the five recombinant antigens varied steeply between the individual mice, in particular for IgM. The level of vaccine-induced circulating anti-5cVAC-IgG in the five analyzed mice was in a rather similar order (with a discrepancy of ± 50% maximal) before and after challenge infection. The highest reliable antibody responses in the samples of the individual mice (Figure 6) and in the pool plasma (Figure 7) were obtained against the antigens PmpD and Ctad1, and partially also against PmpH. The pool serum of vaccinated and challenged mice remained positive against the 5cVAC mix (and above the OD value of the negative control) at a dilution of approximately 1:100,000 or higher (Figure 7). Within the sensitivity of the assays, (almost) no antibodies against PmpA could be detected.

Most likely, due to limited sensitivity of an ELISA utilizing homogenate of Chlamydia-infected cells, we could not verify directly the presence of antibodies raised by adjuvanted 5cVAC against ‘endogenous’ chlamydial antigens (data not shown). However, the IgA and IgG responses against the recombinant antigens seven days after challenge infection were mostly higher (or equal) to those found in the same animal directly before challenge infection (Figure 6). That indicates that the recombinant proteins of 5cVAC had indeed primed the immune system for the corresponding endogenous chlamydial proteins.

### 3.6. Intranasal Vaccination with c-di-AMP-Adjuvanted Two-Component 2cVAC Is Still Effective against C.tr. E, but with A Smaller Degree of Protection as Compared to 5cVAC

After vaccination with 5cVAC, PmpD and Ctad1 were the two antigens that led to the most robust specific IgA and IgG response (Figure 6 and Figure 7). Therefore, they were selected for a vaccination experiment applying i.n. an equimolar mix of these two proteins, only (2cVAC). Compared to non-vaccinated control mice, starting on the fourth day of C.tr. E challenge infection, pretreatment with 2cVAC resulted in similar regain in body weight as with 5cVAC in a former, otherwise identical experiment (Figure 8a). However, the decrease in the clinical score was less pronounced after vaccination with adjuvanted 2cVAC.

On days 5 and 6 p.i., the clinical score of 2cVAC-pretreated mice was smaller than that of mice which had only received buffer or adjuvant, but higher than that of 5cVAC-pretreated animals (Figure 8b). In accordance with that, the amount of infectious Chlamydia in the lung of 2cVAC-treated mice was lower than in the negative controls, but higher than in the 5cVAC-pretreated animals (Figure 8c). Moreover, the levels of MPO and IFN-γ in the lung homogenate of 2cVAC-vaccinated mice were lower than those of non-vaccinated mice. In contrast to 5cVAC-pretreated mice, there was only a similar trend for TNF-α (Figure 8d–f). Thus, the two-component vaccine 2cVAC was still effective against C.tr. E, but achieved a smaller degree of protection as compared to 5cVAC.

### 3.7. Serum-Transfer Demonstrates That Circulating Antibodies Raised by c-di-AMP-Adjuvanted 5cVAC Alone Are Not Protective against C. trachomatis E Lung Infection

To clarify the protective potential of circulating antibodies which are induced by c-di-AMP-adjuvanted 5cVAC after i.n. or s.c. application of the vaccine, respectively, a serum-transfer from vaccinated or non-vaccinated control mice to naïve mice was conducted (Figure 1b, Figure 9 and Figure 10). Two days later, the recipient mice were challenged i.n. with C.tr. E. Moreover, anti-5cVAC antibodies were determined in the transferred pool plasma from the donors as well as in the sera of the individual recipient mice before infection (Figure 9).

Similar amounts of 5cVAC-specific IgM, and total IgG antibodies were present in the pool sera of donor mice that had been vaccinated i.n. or s.c., respectively. Moreover, independently of the route of 5cVAC-application, specific antibodies of the IgG_1_, IgG_2a_ and IgG_2b_ subclass were also detectable in the two pool sera. Thereby, 5cVAC-specific IgG_2a_ seemed to be slightly less elevated in the serum obtained after i.n. vaccination; whereas IgG_1_- and IgG_2b_-levels appeared to be similarly high (Figure 9). As expected, 5cVAC-specific IgA antibodies were only present in the transferred pool serum of donor mice that had been vaccinated by the i.n. route, i.e., via the mucosa (Figure 9, upper left panel). Remarkably, after C.tr. E challenge infection of the serum-recipients, the comparison of body weight, clinical sore, chlamydial load in the lung, MPO, TNF-α, or IFN-γ (Figure 10a–f) between the different groups did not show any significant effect caused by the transferred two 5cVAC-specific hyper-immune antisera-irrespective of whether these sera were raised by i.n. or by s.c. application of the vaccine.

### 3.8. Results of the Long-Term Protection Study

Following the long-term protection model, five months after the last i.n. booster vaccination, hormone-synchronized female mice differently pretreated with either c-di-AMP-adjuvanted 5cVAC, or adjuvant, or buffer alone, were challenged with 4 × 10^5^ of C.tr. E.

#### 3.8.1. Diminished Weight Loss, Clinical Score, and Higher Bacterial Clearance from the Lung

The daily-assessed body weight (%) (Figure 11a) and the clinical score (Figure 11b) showed an improved course of disease of the i.n. 5cVAC-vaccinated mice, starting from day 4 or 5 after challenge infection. Corresponding to that, five months after the last i.n. application of adjuvanted 5cVAC, the bacterial load in lung homogenate was significantly smaller 7 days after C.tr. E challenge infection as compared to control mice that had been pretreated with c-di-AMP only, or buffer (Figure 11c). No significant decrease of MPO, TNF-α, or IFN-γ occurred due to vaccination (Figure 11d–f). Nevertheless, these results clearly indicate that i.n. vaccination with c-di-AMP-adjuvanted 5cVAC leads in mice to a certain degree of long-term protection against C.tr.

#### 3.8.2. Vaccine-Specific Humoral Immune Responses in Blood and Cellular Responses in the Spleen Five Months after the Last Intranasal Booster Vaccination with c-di-AMP-Adjuvanted 5cVAC

To gain insight into vaccine-induced specific immune responses in our long-term protection model (Figure 1c), 5cVAC-induced antibody (Figure 12) and cellular (Figure 13) responses were analyzed five months after the last i.n. applications of the adjuvanted vaccine without (i.e., one day before) or seven days after a C.tr. E lung challenge infection, respectively. In this experimental setting, all mice were euthanized on the day of the collection of serum and splenocytes.

Even after that relatively long period, we could still detect 5cVAC-specific IgA, IgM and IgG as well as IgG_1_, IgG_2a_ and IgG_2b_ in blood obtained slightly ahead as well as after challenge infection, although with rather wide variations from mouse to mouse (Figure 12a,b). As expected, without challenge infection, the antibody levels of mice vaccinated i.n. were much lower after four additional months as compared to those determined in pool serum obtained four weeks after the last booster vaccination (Figure 9).

Moreover, splenocytes of vaccinated and control mice from the long-term experiment, harvested either without or seven days after a C.tr. E challenge infection, proliferated in a dose-dependent manner after incubation with 5cVAC. The dose–response curve of splenocytes from vaccinated mice was shifted to lower antigen concentrations in comparison to that of splenocytes from control mice pretreated with adjuvant alone, or buffer (Figure 13a,b).

ELISpot-analysis of the splenocytes demonstrated a 5cVAC-dependent release of IFN-γ, IL-2, IL-4, and IL-17 after restimulation (Figure 13c,d). For unknown reasons, the background release of IL-17 caused by adjuvant or buffer was relatively high in the splenocytes of mice that were not challenge infected. Nevertheless, the number of IL-17 positive spots was significantly higher after re-stimulation with 5cVAC (Figure 13c). The determined number of spot forming units was similar or higher 7 days after C.tr. E challenge infection (Figure 13d); the number of IFN-γ, IL-2, IL-4, and in particular IL-17 producing splenocytes from mice that had been vaccinated i.n. with c-di-AMP-adjuvanted 5cVAC increased drastically due to re-stimulation with 5cVAC. In spleens of the control mice, such IL-2, IL-4, and IL-17 producing cells were almost absent. Most likely, due to the ongoing challenge infection, there was only a slightly higher background in IFN-γ spot forming units (yet, with a three-times higher response of splenocytes from vaccinated mice).

Therefore, we determined only from splenocytes of challenged mice double or triple positive CD4^+^ or CD8^+^ T cells by FACS-analysis (Figure 13e,f). Indeed, only after re-stimulation with 5cVAC, similar elevated numbers of (IFN-γ/TNF-α/IL-2)-triple-positive, (IFN-γ/TNF-α)-double-positive, to a smaller extend also (IFN-γ/IL-2)-double-positive, and IFN-γ-single-positive CD4^+^ T cells could be detected (Figure 13e). Moreover, there was also an increase in mainly (IFN-γ/TNF-α) and partially also (IFN-γ/IL-2)-double positive CD8^+^ T cells from vaccinated mice after re-stimulation with 5cVAC. That did not occur in T cells from control mice (Figure 13f).

These findings show that vaccination with c-di-AMP-adjuvanted 5cVAC causes long-lasting antigen-specific antibody as well as T cell responses, which correlate with protection against C.tr.

## 4. Discussion

The vaginal mouse infection model with the strict mouse pathogen *C. muridarum* leads to histopathological changes of the upper genital tract and infertility, and is thus, rather close to the human disease the vaccine is intended for. However, with an observation period of up to nine weeks post (challenge) infection, it is rather lengthy. In addition, vaginal application of human C.tr. strains does not cause ascending genital infection with the complications feared in women, but only vaginal shedding for several weeks. Transcervical-intrauterine application of C.tr. as performed by others [21]**,** is technically demanding and only partially physiological. Of course, these urogenital models are nevertheless extremely valuable in C.tr. vaccine research.

Usually, C.tr. can cause pneumonia in human newborns only [8]. Nevertheless, for practical reasons, as demonstrated here, our mouse C.tr. lung infection model [40,41] complements, and in some regards even surpasses the valuable confirmatory urogenital model with all its pros and cons, as a relatively fast and highly quantifiable screening method. Moreover, we had to consider that the identities of the recombinant antigens derived from C.tr. E are, compared to other C.tr. serovars, lower for *C. muridarum* (67–82%; Table 1). Therefore, we chose a model which does not depend on this mouse pathogen and an uncertain degree of cross species protection of 5cVAC. In this work, the C.tr. pneumonia model was also instrumental to determine in detail the efficacy of our new adjuvanted vaccine candidate against various serovars (i.e., broadness of the elicited response and potential universal character of the vaccine candidate), to compare the five (5cVAC) with a two component (2cVAC) vaccine, to address long-term protection, and to characterize vaccine-induced specific humoral and cellular immune responses.

An ideal vaccine should increase the chances of cross-protection between various C.tr. serovars in respiratory and genital (and even ocular) infection. Additionally, it should diminish the risk of escape mutants of the bacterium under selective pressure of population-wide vaccination. Hence, a multi-subunit vaccine, which combines several essential targets, appeared highly advantageous.

The rational for selection of the five chlamydial antigens of C.tr. serovar E contained in 5cVAC was additionally the following: (a) as described above, literature and our own data [23,24,30,32] indicate Pmp’s as promising antigens; (b) these proteins as well as Ctad1 are located on the surface of infectious EBs [33,51]; (c) they are participating in adhesion and binding of C.tr. E to the host cell [24,33], yet, the functional importance of these surface proteins for binding might vary depending on serovar, infected host cell type or tissue; and (d) there is high sequence identity between the Pmp and Ctad1 proteins across C.tr. serovars (E versus L2 or A: 92–100%), while it is lower to other species (C.tr. E versus *C. muridarum*: 67–82%), thus increasing chances for broad cross-serovar protection (see also Table 1). If the humoral response played a role in 5cVAC-induced protection, it seemed likely that induced antibodies bound to several of these antigens on the surface of EBs might neutralize chlamydial binding and diminish infection to the highest extent. With that said, of course, one could not be sure how far specific cellular immune responses might be alternatively or additionally relevant in vaccine-induced defense against the intracellular pathogen after its uptake, and whether antibodies might represent a minor contributor or just a correlate of protection, as our passive serum transfer studies seem to suggest (see below).

Intranasal application of c-di-AMP-adjuvanted 5cVAC was highly protective against C.tr. serovar E: After lung challenge infection, compared to controls the bacterial load was reduced by 2 log10 levels. Loss of body weight and clinical score, as well as the levels of the granulocyte marker MPO and the cytokines IFN-γ and TNF-α in the lung were significantly diminished. The degree of determined protection and the kinetics were similar as observed in a C.tr. E/C.tr. E reinfection experiment under similar conditions (unpublished data). Vaccinated as well as re-infected mice become sick for 3 to 4 days after i.n. challenge before they recover within a few days. The similar course suggests maximal achievable protection by our new vaccine. Importantly, after i.n. application of adjuvanted 5cVAC, effective cross-protection developed against serovars D and L2, and even the ocular serovar A indicating a broad vaccine-induced protection across various C.tr. serovars. The significantly more favorable development of body weight and clinical score and the diminished chlamydial load in lung compared to control mice prove that vaccine-induced protection against C.tr. lasts at least five months. Remaining protection seems to be lower five months after the last i.n. booster application with adjuvanted 5cVAC. Thus, for instance, there were no 5cVAC-dependent changes in the level of MPO, IFN-γ or TNF-α in lung homogenate, and the bacterial load decreased only by approximately 80% as compared to 99%. Yet, a direct comparison of the results achieved in this long-term protection experiment with those achieved in the short-term protection experiment is only possible to a limited extent, because the amount of C.tr. E in the challenge infections was different. It had to be reduced by a factor of >3 in the 31-week-old, more sensitive mice (see also M&M and Appendix A). In this regard, it is critical to consider the normal life-span of mice, the age of the mice included in this experiment at the time of the challenge, and the potential effects of an ongoing immune senescence process.

The results of our Ig-ELISAs were highly suggestive for a functional significance of circulating antigen-specific antibodies induced after i.n. 5cVAC application. The highest reliable (of the rather heterogeneous) antibody responses in 5cVAC i.n. treated mice were obtained against the antigens PmpD and Ctad1, and partially also against PmpH.

Adjuvanted 2cVAC—i.e., an equimolar mix of recombinant PmpD and Ctad1—was still effective against C.tr. E. Yet, according to the clinical score as well as chlamydial clearance and MPO, IFN-γ, and TNF-α in the lung homogenate, 2cVAC was less effective than 5cVAC. One can draw the conclusions (a) that PmpD and/or Ctad1 are highly protective and (b) that at least one of the other three antigens contained in 5cVAC is additionally inducing protection, emphasizing the importance of a multi-subunit vaccine. That is of even more concern, as one cannot exclude that the success of the vaccine-induced defense might also depend on the respective C.tr. serovar and the infected organ. Furthermore, a multi-subunit vaccine reduces the risk of immune escape resulting from either pathogen evolution or vaccination-derived selective pressure.

This risk might be much higher in case of a ‘single-antigen’ MOMP-based vaccine. In C.tr. vaccination studies, MOMP plays a prominent role [6,16,19]. According to the overview by Phillips et al. from 2019 [5]: “To date, the MOMP has emerged as the most suitable substitute for whole cell targets and its delivery as a combined systemic and mucosal vaccine is most effective.” Yet, as reviewed in [19], MOMP induces C.tr. serovar- or at least serogroup-specific responses. Furthermore, against *C. muridarum*, nMOMP purified (in small amounts) from the chlamydial EBs was more protective than purified recombinant rMOMP [52,53,54]-limiting for practical reasons its use as vaccine. Presently, the most suitable and promising substitute for a chlamydial whole cell target is probably a complex recombinant protein vaccine combining the variable domain VD4 and surrounding constant immunogenic regions of MOMP from different serovars, in order to cover the most frequent ones, combining s.c. and i.n. administration [55]. A promising phase-1-clinical trial has been recently finished [56]. However, without loss of function the MOMP gene *ompA* can mutate and is subject to immune selective pressure and recombination, as demonstrated by the existing various C.tr. genotypes and serovars [57]. The identification of a novel, hybrid C.tr. genomic mosaic L2b/D-Da strain causing an outbreak of LGV indicates also this ability of C.tr. for genetic variation [58]. Population-wide vaccination with parts of MOMP—if successful—might raise such pressure leading to the selection of existing C.tr. serovars that are not targeted by the vaccine, and of new C.tr. MOMP mutants or variants who escape the defense caused by vaccine-induced antibodies or T cells.

Interestingly, after vaccination with shorter recombinant fragments from all nine C.tr. E Pmps combined with CpG-1826 and Montanide ISA 720 as adjuvants by 2× the i.m. plus 1× the s.c. route, BALB/c mice vaccinated with PmpC, G, or H were best protected against an i.n. *C. muridarum* challenge infection [30]. In that quite different experimental setting, the IgG serum titer against PmpG was the highest, PmpH was on the third position, PmpD on the fifth and PmpA second to last. Thus, in that [30] and our study, only the low humoral responses to PmpA are in total accordance.

It is important to note, however, that high immunogenicity or immuno-dominance of an antigen does not mean that it must participate in the observed defense. Antibodies might be just a ‘biomarker’ correlating with protection rather than a main effector mechanism for the immune protection achieved. In fact, after vaccination with adjuvanted 5cVAC, circulating antibodies play, if at all, only a minor role, as most convincingly demonstrated by absent protection against C.tr. E after transfer of hyper-immune serum.

The rather similar measured values of 5cVAC-specific antibodies in the blood of individual recipients (collected on the day between serum-transfer and i.n. C.tr. E-challenge infection) within each treatment group demonstrate the reliable performance of the serum transfer experiment. The amount of C.tr.-specific antibodies in the recipients was approximately 10% (1:7 to 1:20) of the amount found in the administered pool serum. The measured factors are pretty close to the predicted dilution factor of approximately 1:6. This prediction takes into account only the addition of the known volume of transferred serum (2 × 170 µL) to the naïve recipient’s (according to body weight) estimated total blood volume, but e.g., not redistribution into other body fluids. Serum transfer did not lead to any protection in the recipients. Thus, approximately 1:10-diluted vaccine-induced circulating antibodies alone are not protective against C.tr. E.

This view is further supported by negative preliminary results obtained in a flow cytometric neutralization assay with HeLa or Syrian hamster kidney (HaK) cells after preincubation of EBs from C.tr. E with the hyper-immune serum (data not shown). Delayed improvement of body weight and clinical score after an unchanged onset of disease argue also against a main protective role of antibodies because one might expect that, if neutralizing EBs, they would prevent host cell infection and thus, already affect the start of challenge infection. Finally, in the plasma of the fifth of the individually analyzed animals, 5cVAC-specific IgA, IgM or IgG were hardly detectable after vaccination. Yet, an approximately 50-fold (compared to the mean of other mice smaller) decrease of the bacterial load still occurred (data not shown). Of course, one cannot completely exclude that small amounts of circulating antibodies below the detection limit of our Ig-ELISA were already interfering in infection of this mouse.

Subcutaneous application of c-di-AMP-adjuvanted 5cVAC was not protective in C.tr. E challenge. Only the i.n., i.e., mucosal application of our new vaccine protected against chlamydial lung infection. Our findings comparing the antibody responses after vaccination by the two different routes exclude that smaller amounts of circulating 5cVAC-specific IgM, IgG or IgG_1_, IgG_2a_, and IgG_2b_ are responsible for that. However, our results—in particular those of the serum-transfer experiment—do not rule out augmentation of T cell responses by circulating antibodies, or a critical role of mucosal, secretory IgA.

In a C.tr. study of Stary et al. [21], mice were vaccinated with UV-inactivated EBs alone or with commonly used adjuvants. Surprisingly, more Chlamydia were found after genital challenge infection in the uterus of vaccinated animals as compared to controls. Most likely, tolerance was induced by regulatory T cells-explaining negative outcomes of trials based on whole cell EBs in the 1960s. In parallel, UV-inactivated EBs covalently linked to a potent TLR7/8 agonist and complexed with nano charge-switching synthetic adjuvant particles were also applied. Intriguingly, similar to our new vaccine, s.c. vaccination was not protective. However, genital as well as i.n. application resulted via mucosal cross-protection in a decreased chlamydial load in that organ. After mucosal application of the adjuvanted vaccine, CD4^+^ effector T cells migrate to the uterus and establish tissue-resident memory cells. These cells are reactivated upon genital challenge triggering migration of circulating memory T cells into the infected organ. In a recall response, they release cytokines for pathogen clearance, thereby blocking infection at an early stage. According to the Stary study, both migration waves are required for optimal defense against Chlamydia. In contrast to oral vaccination, i.n. vaccination leads to efficient responses not only to local nasopharynx- and bronchus-associated lymphoid tissue (NALT/BALT), but also at distant mucosal territories, such as in vaginal secretions of the genitourinary tract [59]. Moreover, mucosal sites function as a common system-wide organ, which act as an interface between the physical environment and the host mucosal defenses [60]. On this background [21,58,59], one can draw the conclusions that adjuvant and route of application play critical roles for the success of a vaccine against the primarily mucosa-infecting Chlamydia. Furthermore, a similar immunological mechanism as described above might also be involved in our case. Thus, one can speculate that after i.n. application cross-mucosal protection including the urogenital tract might also be achieved with 5cVAC, if combined with the right adjuvant.

It seems reasonable to speculate that 5cVAC-induced cellular immune responses of CD4^+^, CD8^+^, memory, tissue-resident, regulatory, and other T cell subsets, might be more important than antibodies for the observed success of i.n. vaccination. An unambiguous clarification of this issue is beyond the scope of this work and will need to be solved by further studies. However, both the IgG subtypes of the raised antigen-specific antibodies in plasma or serum and the ELISpot-analysis of splenocytes after restimulation with 5cVAC strongly suggest the induction of a mixed Th1/Th2/Th17 immune response. The remarkably high stimulation of antigen-specific CD4^+^ T cells still 5 months after i.n. vaccination underscores the potency of c-di-AMP as mucosal adjuvant.

In addition to buffer, i.n. application of c-di-AMP alone was included as negative control in most experiments, to rule out any potential short-term protective effect as a result of c-di-AMP-mediated immune stimulation. As expected, it did not have any effect, as demonstrated in the short-term protection model for C.tr. E or D on body weight, clinical score (Figure 2a1–b2), bacterial load and MPO (Figure 3a1–b2), or TNF-α and IFN-γ (Figure 4a1–b2). The adjuvant did also not modify the same parameters in the 2cVAC short-term experiment (Figure 8a–f) and the long-term experiment (Figure 11) using C.tr. E for challenge. In addition, there was also no visible effect of the adjuvant alone in non-vaccinated mice on the levels of antigen-specific antibodies (Figure 9) or the responses of their splenocytes (Figure 13). Based on these multiple identical results and for practical reasons, the adjuvant control (in addition to the buffer control) was not included in the analysis of the observed protective effect of adjuvanted 5cVAC in challenge infection with C.tr. A in the short-term model (Figure 2d1,d2, Figure 3d1,d2 and Figure 4d1,d2), and partially also with C.tr. L2 (Figure 4c1,c2). However, it seems highly unlikely, that a non-specific effect of c-di-AMP alone should only take place after challenge infection with serovars, other than D or E.

The adjuvancity of c-di-AMP is explained by the stimulation of STING (stimulator of interferon genes), which in turn leads to a signal transduction cascade promoting expression of type I interferons and TNF-α. The proof of concept for vaccine candidates against several other pathogens (e.g., influenza virus, hepatitis viruses, *Trypanosoma cruzi*, *Streptococcus pyogenes*, etc.) causing mucosal and non-mucosal infections has been already provided in different experimental animal models [36,37,38,39] and some of them are currently going into clinical development. Most likely due to incorporation of c-di-AMP in the i.n. 5cVAC vaccine, strength and quality of the antigen-specific CD4^+^ and CD8^+^ T cell responses were also augmented here, as indicated by the presence of bi- and trifunctional CD4^+^ and CD8^+^ T cells. These cytokine producers were dominated by cells double positive for IFN-γ and TNF-α, which is in agreement with other studies using c-di-AMP as adjuvant. Such multifunctional CD4^+^ T cells have been demonstrated to be important during C.tr. infection, e.g., in pigs. Käser and co-workers revealed that C.tr. infections result in the induction of CD4^+^ T cells that are either IFN-γ-single, or (IFN-γ/TNF-α)-double cytokine-producing T-helper 1 cells [61]. Interestingly, IL-17-producing CD4^+^ T cells were rare or completely absent. One might expect that several overlapping and difficult to dissect cellular or humoral immune mechanisms might contribute to 5cVAC-induced protection against C.tr.

Taken together, we obtained in mice with our new adjuvanted multi-subunit vaccine highly promising results in lung C.tr. challenge infection, and gained insights into the vaccine-induced immune response. The three main reasons of our success might be: (A) we used in 5cVAC longer protein fragments of the extracellular Pmp PDs than others combining them with full length Ctad1 as antigens; (B) with c-di-AMP, a different mucosal adjuvant was chosen, which primarily stimulates dendritic cells and has been shown to be capable of promoting both, humoral and broad cellular immune responses [59,62]; and (C) the adjuvanted protective vaccine was administered i.n., i.e., to the mucosa.

Nevertheless, there is still a long way to go. (1) The composition and ration of the subunits should be optimized. (2) Our results suggest that 5cVAC-induced protection diminishes five months after the third application of the adjuvanted vaccine. We want to investigate how far the mode and timing of application influences duration of protection, e.g., combining mucosal and non-mucosal application of 5cVAC, or using an additional booster application. (3) It is likely, that c-di-AMP-adjuvanted 5cVAC is also protective in genital mouse infection, and that this protection includes several C.tr. serovars. Based on our results, the next major step in the development of our vaccine will be to switch to a urogenital mouse model. Now, that we know that effective cross-serovar protection is inducible with i.n. adjuvanted 5cVAC, it will be critical to assess how far mucosal cross-protection is achievable in the urogenital tract after i.n. application of 5cVAC, as described for other antigens [21]. With c-di-AMP as adjuvant, this might actually be the case, as demonstrated already for other pathogens. So far at least, c-di-AMP lived up to most of our expectations, e.g., the induction 5cVAC-specific antibody and mixed Th1/Th2/Th17 T cell responses, as well as CD8 cytotoxic T cells, and relevant induced protection [36,37,38,39]. Intranasal or sublingual administration should be able to promote such responses in the genitourinary tracts, as previously demonstrated for HIV and HSV antigens (unpublished data), however, vaginal administration of adjuvanted 5cVAC might be also a feasible alternative. With such experiments in mind, we will check before, how far i.n. adjuvanted 5cVAC cross-protects against *C. muridarum* in mouse lung infection. In any case, we can use transcervical-intrauterine application of C.tr. to investigate in longer lasting, but more focused experiments vaccine-induced protection in the urogenital tract. (4) Our observation of the induced immune response in the lung infection model as well as the analytic transfer experiment of serum (with circulating antibodies) provide insight into 5cVAC-induced immune responses. On that basis, one can develop hypotheses about the nature of induced protection. Yet, future experiments are also required to confirm the presumed protective immune mechanisms by challenge of 5cVAC-vaccinated mice who exhibit defects in parts of their immune system—e.g., K/O mice without functional B cells and antibodies including IgA, or by challenge infections of vaccinated CD8^+^ or CD4^+^ T cell-depleted animals. However, the fact that deficits in the B cell compartment can also affect cellular immunity [63], might render final analysis difficult.

## 5. Conclusions

Our results show that repetitive i.n. vaccination with 5cVAC - consisting of 4 recombinant Pmp proteins and Ctad1 derived from C.tr. E combined with the adjuvant c-di-AMP - protects mice efficiently against an experimental lung challenge infection with C. tr. E. Moreover, cross-serovar protection was induced as demonstrated for 2 additional urogenital and 1 trachoma serovar. Experiment using 2cVAC showed that at least one of the two contained antigens (PmpD and Ctad1) and at least one of the three not-contained antigens (PmpA, PmpG, and PmpH) is inducing protection. This result emphasizes the importance of a multi-subunit vaccine. In addition, such a vaccine should be less prone for selection of immuno-evasive C.tr. mutants. Protection, although probably weakened, was still detectable 5 months after the last i.n. application of 5cVAC. The importance of mucosal application is indicated by the lack of protection observed after s.c. application of our vaccine. High, but varying amounts of antigen-specific IgA, IgM, and IgG are induced by i.n. vaccination. Yet, serum-transfer experiments suggest that circulating antibodies play a minor role in vaccine-induced protection. Most likely, the observed antigen-specific T cell responses are functionally more important in case of our vaccine. While still subject to optimization and testing in mouse urogenital and other animal models, the results obtained here are highly promising. We hope that, in the long run, our new vaccine might help to prevent the feared consequences of human C.tr. urogenital infections, in particular infertility, as well as blindness due to trachoma.

## 6. Patents

A patent application for the c-di-AMP-adjuvanted vaccine 5cVAC naming A.K., R.L., J.H.H., S.W., T.E., and C.A.G. as inventors has been filed (EP2115349.6; not published, yet). C.A.G. and T.E. are named as inventors in patents covering the use of c-di-AMP as vaccine adjuvant (PCT/EP 2006010693), as well as neonatal adjuvant (EP 19193982).

## Figures and Tables

**Figure 1 vaccines-09-00609-f001:**
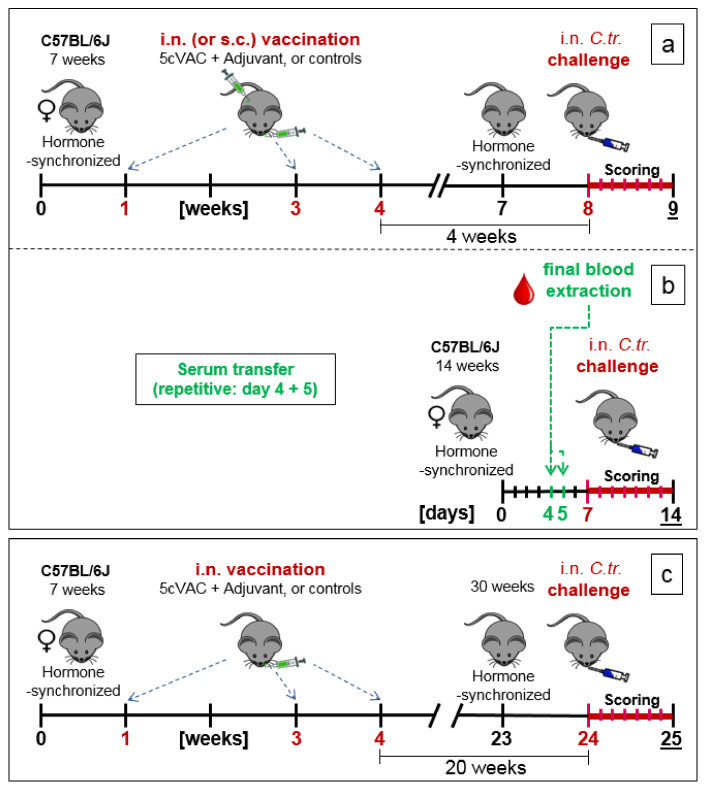
Experimental setup of the three vaccination trials using the *Chlamydia trachomatis* lung challenge infection models: short-term (**a**), serum transfer-short-term (**b**), and long-term protection (**c**). Experimental setup of short-term protection model (**a**) 1, 3, and 4 weeks after hormone-induced synchronization, eight-week-old female C57BL/6J mice were repetitively treated with an identical volume of c-di-AMP-adjuvanted antigens, or adjuvant, or buffer, respectively. The administered 5cVAC contains an equimolar mix of 5 (and 2cVAC of 2) recombinant antigens derived from serovar E of *C. trachomatis* (C.tr. E). Four weeks after the last booster vaccination, the 15-week-old hormone-synchronized mice were i.n. challenge-infected with different strains/serovars. Under close monitoring and using humane endpoint defining criteria, body weight and clinical score were determined daily. On day 7 of challenge infection, all mice were painlessly sacrificed for further analysis of their blood and lung tissue. Serum transfer-short-term protection model (**b**) four weeks after administration of the third dose of adjuvanted 5cVAC or adjuvant, or buffer, respectively, serum of healthy donor mice was collected by heart puncture. The pooled sera were transferred to 14- to 15-week-old healthy, hormone-synchronized female mice. Then, lung challenge infection was performed with the identical amount of C.tr. E as used in (**a**). Long-term protection (**c**) vaccination with adjuvanted 5cVAC was conducted as described in (**a**). However, the period between the last i.n. administration of the vaccine and challenge infection with C.tr. E was prolonged from 4 to 20 weeks, and a smaller amount of C.tr. E was applied to the then older and more sensitive mice.

**Figure 2 vaccines-09-00609-f002:**
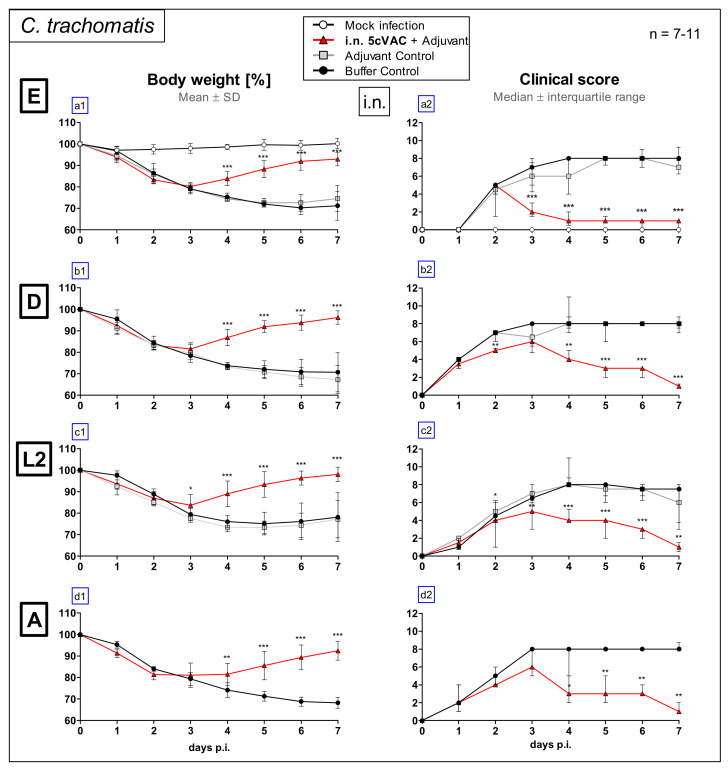
Loss of body weight and clinical score in lung challenge infection with *C. trachomatis* serovar E, D, L2, or A (marked by boxed letters to the left) after i.n. vaccination with c-di-AMP-adjuvanted 5cVAC. In the short-term protection model (see Section 2 and Figure 1a), mice were either pre-treated with c-di-AMP-adjuvanted 5cVAC, the adjuvant alone, or buffer, as indicated. *N* = 7–11 differently pretreated animals per group for the C.tr. genital serovars E, D, and L2, and 7 mice per group for C.tr. A, respectively, were infected i.n. with the different serovars, or mock-infected (*n* = 8). Day 0 post infection (p.i.) in this graph corresponds to week 8 in Figure 1a. During the following seven days, body weight (%) (left panels (**a1**–**d1**), mean ± standard deviation) and clinical score (right panels (**a2**–**d2**), median ± interquartile range) were assessed daily. The performed statistical analysis is described in Section 2. *,**,*** indicate statistical significances between the 5cVAC group (red) and the buffer control group (black) with *p* < 0.05, <0.01, and <0.001, respectively. The exact size of each group can be found in Appendix A.

**Figure 3 vaccines-09-00609-f003:**
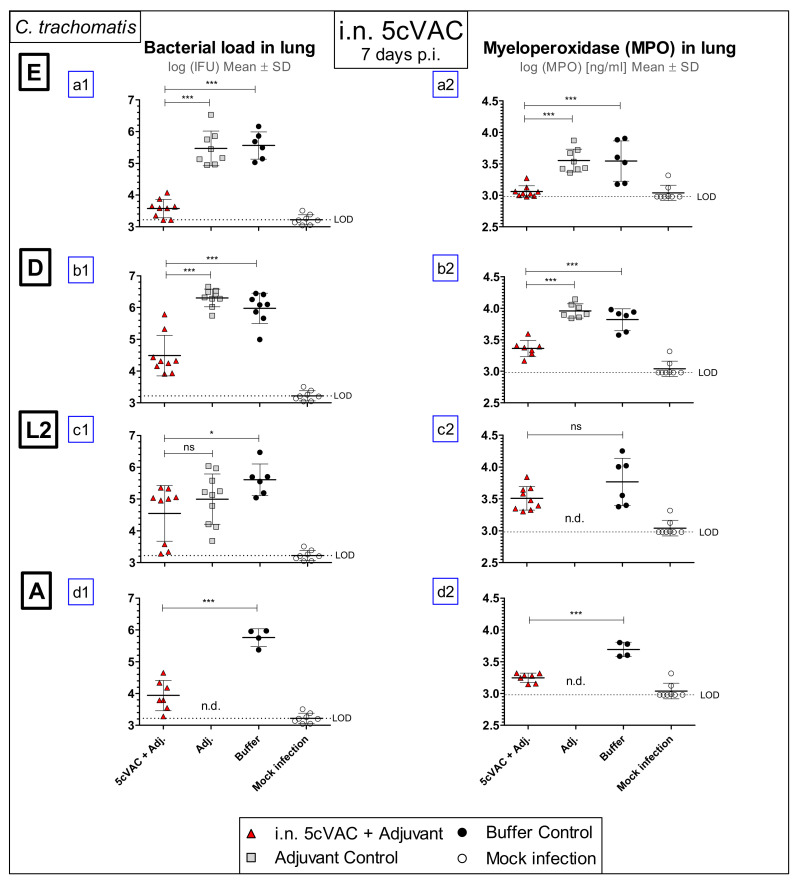
Bacterial load and granulocyte marker MPO in the mouse lung on day 7 of challenge infection with various serovars of *C. trachomatis* after i.n. vaccination with adjuvanted 5cVAC. In the short-term protection model (Figure 1a), mice were either pre-treated with c-di-AMP-adjuvanted 5cVAC, the adjuvant alone, or buffer, as indicated. The mice were challenged with C.tr. serovar E, D, L2, or A (marked by boxed letters to the left) to check for cross-serovar protection (**a1**–**d1**, **a2**–**d2**). On day 7 of chlamydial challenge (i.e., week 8 in Figure 1a), the amount of viable, infectious Chlamydia (left panels, mean ± standard deviation of log10 (IFU)) and the level of MPO (right panels, mean ± standard deviation of log10)) were determined in lung homogenate of the pretreated animals surviving that period (*n* = 6–10 per group for serovar E, D and L2, *n* = 4–7 for serovar A—as indicated by the number of symbols in the figure). Mock infection: n = 8. LOD = limit of detection; n.d. = not determined; ns = not significant. The performed statistical analysis is described in Section 2. * and *** indicate statistical significances with *p* < 0.05 and <0.001, respectively. The exact size of each group can be found in Appendix A.

**Figure 4 vaccines-09-00609-f004:**
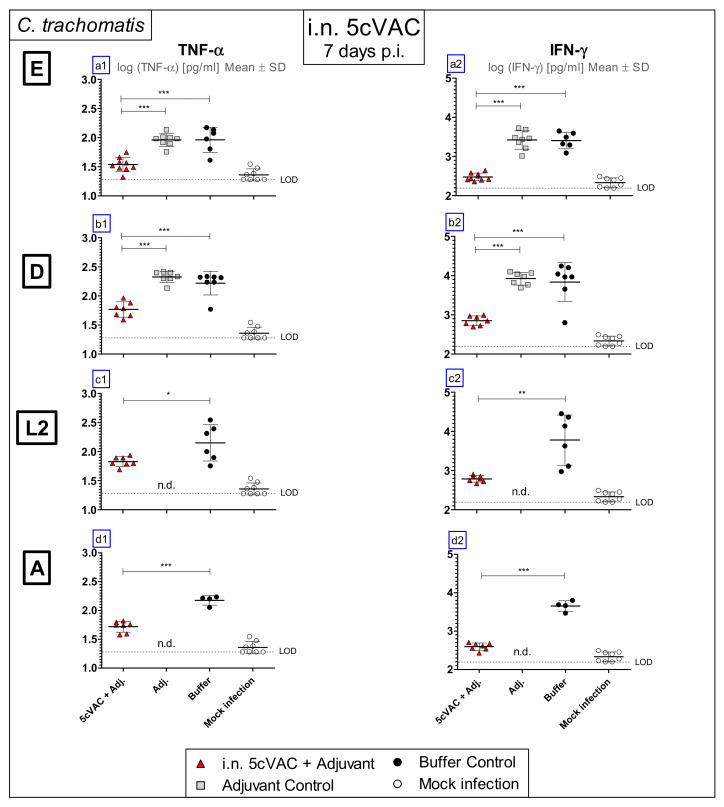
TNF-α and IFN-γ in lung homogenate on day 7 of challenge infection with various serovars of *C. trachomatis* (marked by boxed letters to the left) in the short-term protection model after i.n. vaccination with adjuvanted 5cVAC. Mice were either pre-treated with c-di-AMP-adjuvanted 5cVAC, c-di-AMP alone, or buffer, as indicated. The mice were challenged with C.tr. serovar E, D, L2, or A to check for cross-serovar protection (**a1**–**d1**,**a2**–**d2**). On day 7 of chlamydial challenge, TNF-α (left panels, mean ± standard deviation of log10) and IFN-γ (right panels, mean ± standard deviation of log10) were determined in lung homogenate of the differentially pretreated mice surviving that period (*n* = 6–9 per group for serovar E, D and L2, *n* = 4–7 for serovar A – as indicated by the number of symbols in the figure). Mock infection: *n* = 8. LOD = limit of detection, n.d. = not determined. The performed statistical analysis is described in Section 2. *,**,*** indicate statistical significances with *p* < 0.05, <0.01, and <0.001, respectively. The exact size of each group can be found in Appendix A.

**Figure 5 vaccines-09-00609-f005:**
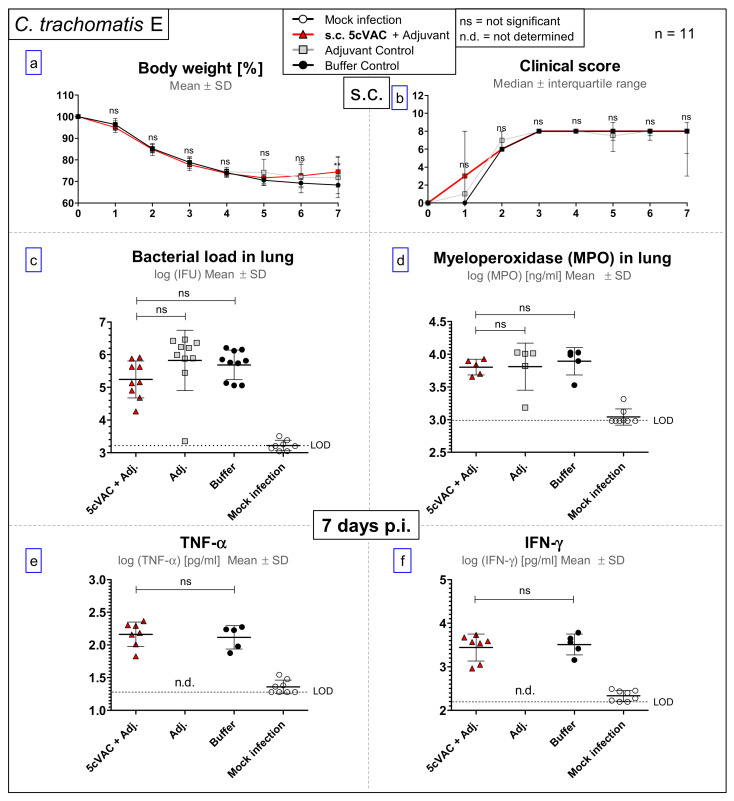
Body weight, clinical score, bacterial load, granulocyte marker MPO, TNF-α, and IFN-γ in lung challenge infection with *C. trachomatis* E in the short-term protection model after s.c. vaccination with c-di-AMP-adjuvanted 5cVAC. In this short-term protection experiment (see Figure 1a), adjuvanted 5cVAC was applied by the s.c. route. Animals pretreated with adjuvant, or buffer, respectively, served as negative controls. Per group, *n* = 11 differently pretreated 15-week-old animals were infected i.n. with the usual amount of C.tr. E (1.3 × 10^6^ IFU), the serovar the antigens in 5cVAC are derived from. On each of the following days, body weight ((**a**), in % as mean ± standard deviation) and clinical score ((**b**), median ± interquartile range) were assessed. On day 7 after i.n. challenge infection, the surviving animals (also indicated by the number of symbols in the figure) were sacrificed for determination of the amount of viable, infectious Chlamydia ((**c**), mean ± standard deviation of log10 (IFU); *n* = 9–10), and the levels (mean ± standard deviation of log10) of MPO ((**d**); *n* = 5), TNF-α ((**e**); *n* = 5–7) and IFN-γ ((**f**); *n* = 5–7) in the lung homogenate. Mock infection: *n* = 8. LOD = limit of detection; n.d. = not determined, ns = not significant. The performed statistical analysis is described in Section 2. ** indicates statistical significance with *p* < 0.01. The exact size of each group can be found in Appendix A.

**Figure 6 vaccines-09-00609-f006:**
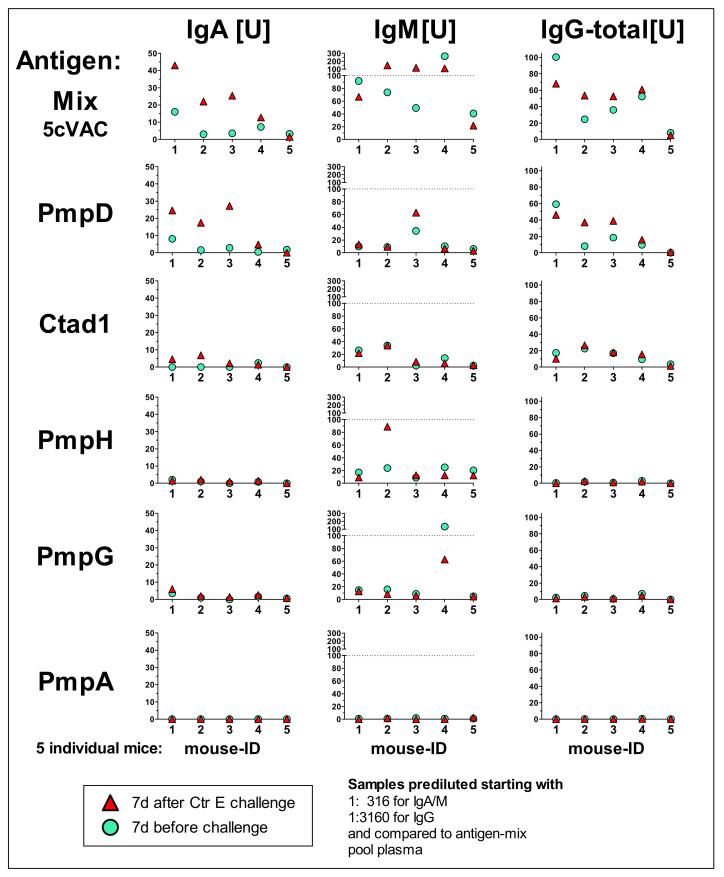
Antibody response profiles towards the 5cVAC-mix or its five components in mouse plasma of five individual mice, seven days before or after *C. trachomatis* E challenge. ELISAs detecting IgA (**left**), IgM (**middle**), and total IgG (**right**) directed against the 5cVAC antigen-mix or the single vaccine components (in the indicated rows) were performed with plasma from individual i.n. 5cVAC-vaccinated mice obtained in the short-term protection model seven days before (green circles) and after (red triangles) C.tr. E (1.3 × 10^6^ IFU) infection. To permit direct comparison of the results, arbitrary units of the different Ig’s [U] were calculated in relation to a standard curve based on pool plasma obtained after 5cVAC application and C.tr. E challenge. The samples had to be prediluted to 1:316 for IgA/M and 1:3160 for IgG, respectively. All corresponding negative controls with plasma of individual non-vaccinated C.tr. E-challenged or non-challenged mice remained below <1 U (data not shown).

**Figure 7 vaccines-09-00609-f007:**
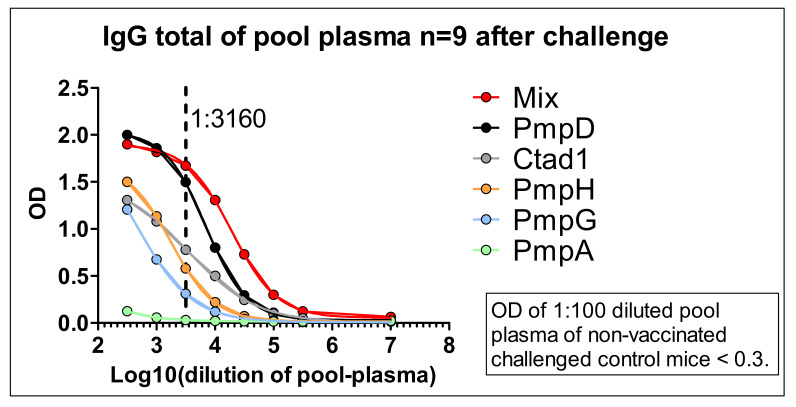
5cVAC-specific IgG-antibodies in pool-plasma obtained from mice vaccinated i.n. with adjuvanted-antigen seven days after challenge infection with *C. trachomatis* E. The amount of IgG directed against the 5cVAC-mix or the individual recombinant antigens, respectively, was determined in pool plasma of *n* = 9 5cVAC-vaccinated and C.tr. E-challenged mice. Due to the larger available volume, dilution could already start at 1:316, and thus, lower antigen-specific IgG-responses are detectable, here. The vertical broken line at 1:3160 indicates the lowest dilution used before in total-IgG analysis of five individual mice (Figure 6). Pool plasma of non-vaccinated control mice that were challenged i.n. for seven days with C.tr. E served as negative control; even just 1:100 diluted, its OD depending on total IgG against the 5cVAC mix remained <0.3 (data not shown).

**Figure 8 vaccines-09-00609-f008:**
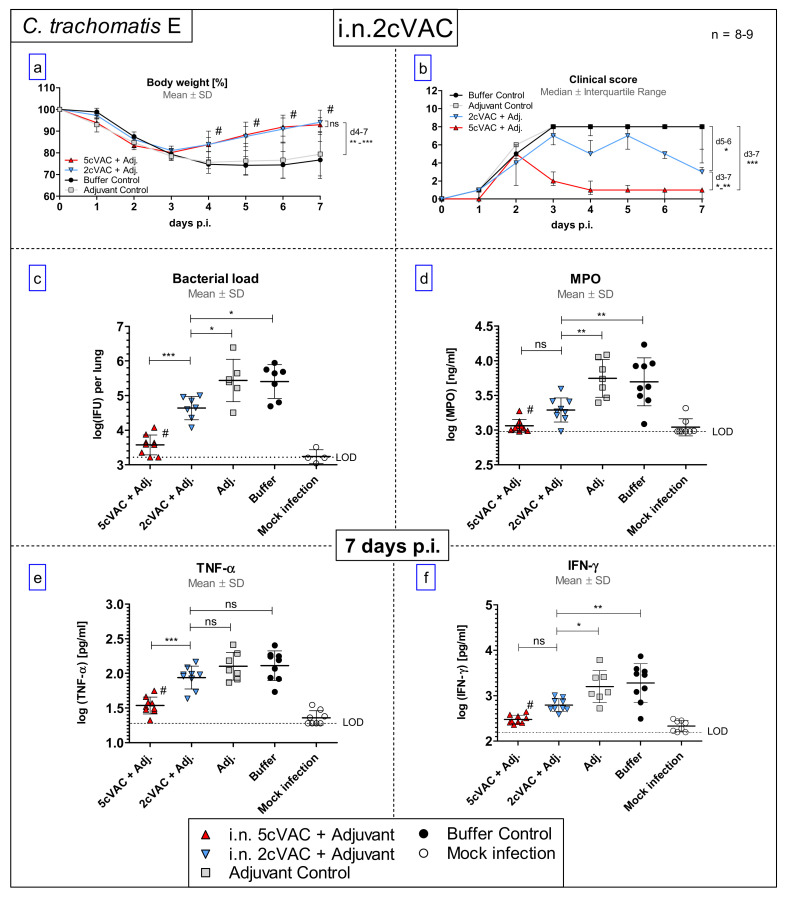
Body weight, clinical score, bacterial load, granulocyte marker MPO, TNF-α, and IFN-γ in lung challenge infection with *C. trachomatis* E in the short-term protection model after i.n. vaccination with c-di-AMP-adjuvanted two-component 2cVAC. In this modified short-term protection model (see Figure 1a), vaccinated mice received only two of the five components contained in 5cVAC (2cVAC: PmpD and Ctad1) plus c-di-AMP by the i.n. route (blue triangles). Animals pretreated with adjuvant, or buffer, respectively, served as negative controls. Per group, *n* = 8–9 differently pretreated 15-week-old animals were infected i.n. with 1.3 × 10^6^ IFU of C.tr. E. On each of the following days, body weight ((**a**), in % as mean ± standard deviation) and clinical score ((**b**), median ± interquartile range) were assessed. On day 7 after i.n. challenge infection, the surviving animals (also indicated by the number of symbols in the figure) were sacrificed for determination of the amount of viable, infectious Chlamydia ((**c**), mean ± standard deviation of log10 (IFU); *n* = 6–9), and the levels (mean ± standard deviation of log10; *n* = 7–9) of MPO (**d**), TNF-α (**e**), and IFN-γ (**f**) in the lung homogenate. Mock infection: *n* = 4 for bacterial load, *n* = 8 for other parameters. For better comparability, data obtained in a previous, otherwise identical experiment with i.n. 5cVAC pretreatment are also depicted (red triangles). LOD = limit of detection; ns = not significant. The performed statistical analysis is described in Section 2. *,**,*** indicate statistical significances with *p* < 0.05, <0.01, and <0.001, respectively. Statistical significance (*p* < 0.05) between the 5cVAC and the adjuvant/buffer control groups is marked with # in the panels (**a**,**c**,**d**–**f**). The exact size of each group can be found in Appendix A.

**Figure 9 vaccines-09-00609-f009:**
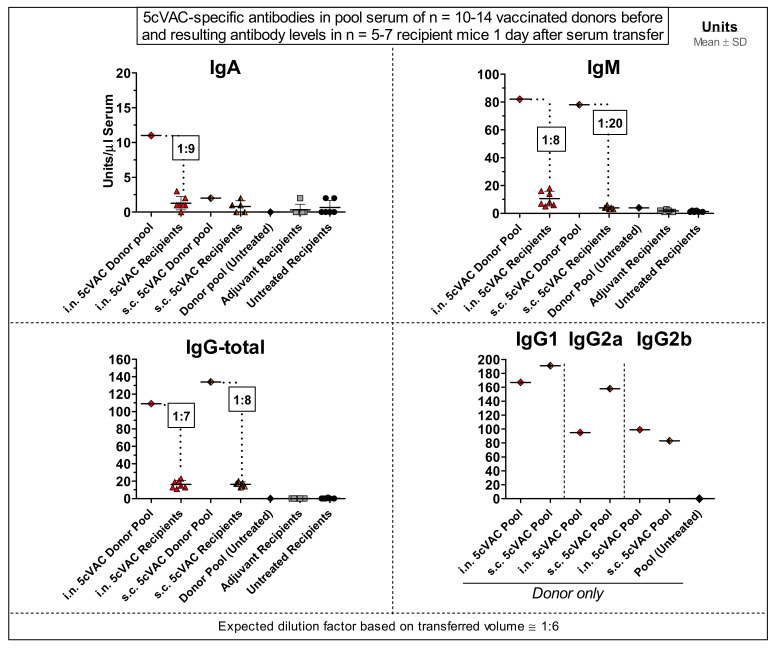
Specific IgA, IgM, total-IgG, and IgG-subtypes in transferred donor pool sera from differentially vaccinated non-infected mice and resulting Ig-levels in healthy recipient mice (before i.n. challenge with *C. trachomatis* E). Hormone-synchronized, non-vaccinated, 15-week-old C57BL/6J (*n* = 5–7 per group) received i.v. pool sera (of 10–14 animals per group) that had been vaccinated i.n. or s.c. with c-di-AMP-adjuvanted 5cVAC (see also Figure 1b). Alternatively, they received pool sera of mice that had been pretreated with c-di-AMP alone (adjuvant recipients), or that were not pretreated at all (untreated recipients). One day after serum administration, a small amount of blood was collected from each recipient to determine resulting individual antibody levels in comparison to the Ig levels found in the corresponding transferred pool serum. The boxes in the graph show the calculated factors of dilution of the 5cVAC-specific antibodies. Anti-5cVAC IgA, IgM, total IgG, as well as IgG_1_, IgG_2a_, and IgG_2b_ were determined as arbitrary units (U) in the transferred pool sera. The IgG-subtypes were only determined in the pool sera from the donors. In this context, “Pool (Untreated)” represents the results combined from the various negative controls. The size of each group can be found in Appendix A.

**Figure 10 vaccines-09-00609-f010:**
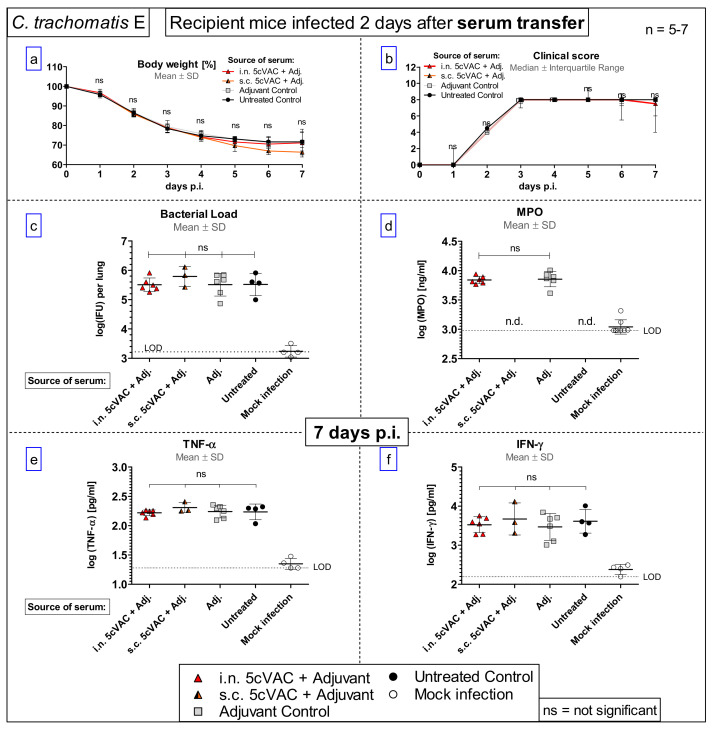
Body weight, clinical score, bacterial load, granulocyte marker MPO, TNF-α, and IFN-γ in recipients of transferred pool sera from vaccinated healthy mice, after i.n. challenge infection with *C. trachomatis* E. Hormone-synchronized, non-vaccinated, 15-week-old C57BL/6J (*n* = 5–7 per group) received i.v. pool sera of 10–14 animals (per group) that had been vaccinated i.n. or s.c. with c-di-AMP-adjuvanted 5cVAC. Other recipients received pool sera of mice that had been pretreated with c-di-AMP alone (Adj.; adjuvant recipients), or that were not pretreated at all (Untreated; untreated recipients). Two days after serum-transfer, the recipients were challenged i.n. with 1.3 × 10^6^ IFU of C.tr. E (see also Figure 1b). Afterwards, body weight ((**a**), in % as mean ± standard deviation) and clinical score ((**b**), median ± interquartile range) were assessed daily. On the seventh day of challenge infection, the animals were sacrificed for determination of the bacterial load of infectious Chlamydia ((**c**), mean ± standard deviation of log10(IFU); *n* = 3–6) and of the levels (mean ± standard deviation of log10) of MPO ((**d**); *n* = 6), TNF-α ((**e**); *n* = 3–6) and IFN-γ ((**f**); *n* = 3–6) in lung homogenate. Mock infection: usually *n* = 4; *n* = 8 for MPO. LOD = limit of detection; n.d. = not determined; ns = not significant. The performed statistical analysis is described in Materials and Methods. The exact size of each group can be found in Appendix A.

**Figure 11 vaccines-09-00609-f011:**
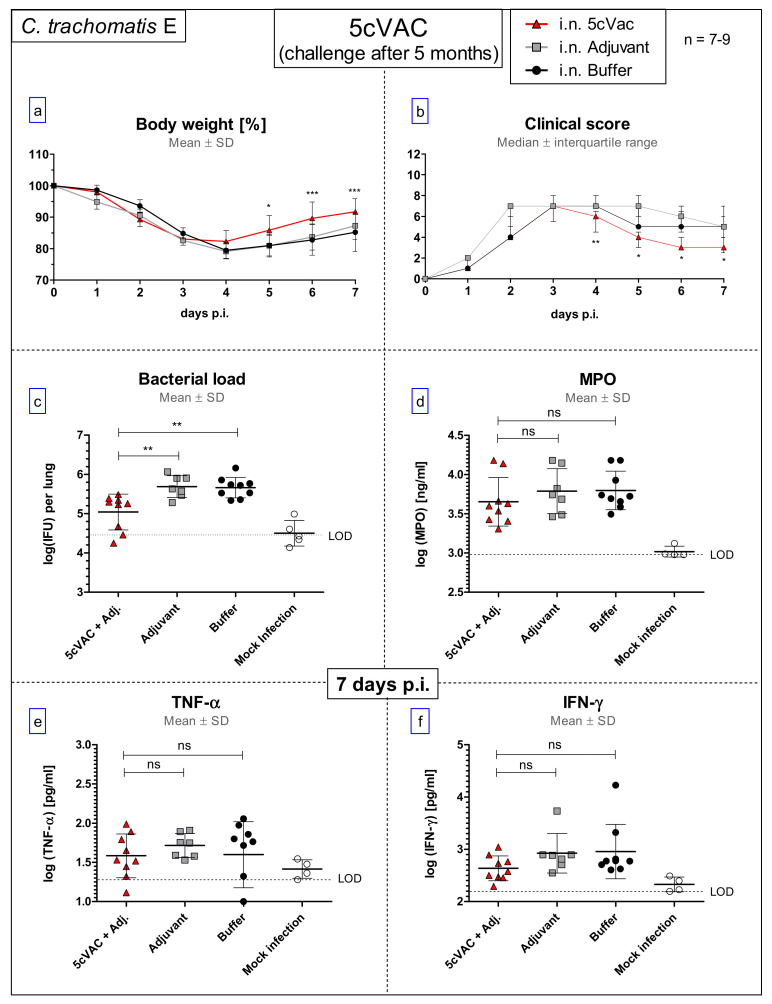
Body weight, clinical score, bacterial load, granulocyte marker MPO, TNF-α, and IFN-γ in lung challenge infection with *C. trachomatis* E in the long-term protection model five months after i.n. vaccination with adjuvanted 5cVAC. Twenty weeks after the last i.n. booster vaccination, *n* = 7–9 hormone-synchronized female mice differently pretreated either with c-di-AMP-adjuvanted 5cVAC, or c-di-AMP or buffer, were challenged with 4 × 10^5^ IFU of C.tr. E (see also Figure 1c) and analyzed. Body weight ((**a**), in % as mean ± standard deviation) and clinical score ((**b**), median ± interquartile range) were assessed daily. On day 7 after challenge infection, the animals were sacrificed for determination of the bacterial load, i.e., the amount of viable, infectious Chlamydia ((**c**), mean ± standard deviation of log10(IFU)), and the levels (mean ± standard deviation of log10) of MPO (**d**), TNF-α (**e**) and IFN-γ (**f**) in the lung homogenate. Mock infection: *n* = 4–5. LOD = limit of detection, ns = not significant. The performed statistical analysis is described in Section 2. *,**,*** indicate statistical significances with *p* < 0.05, <0.01, and <0.001, respectively. The exact size of each group can be found in Appendix A.

**Figure 12 vaccines-09-00609-f012:**
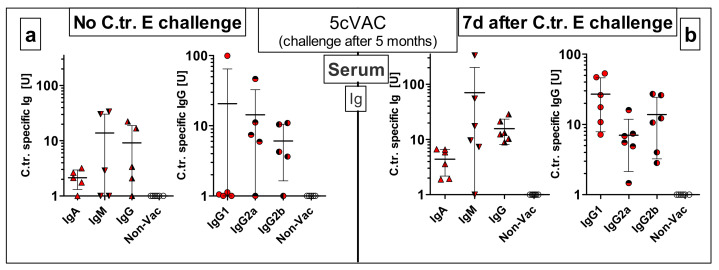
Antigen-specific humoral responses in mice 5 months after the last i.n. booster vaccinations with c-di-AMP-adjuvanted 5cVAC. 5cVAC-specific circulating Ig’s were determined in serum 20 weeks after the last i.n. application of the c-di-AMP-adjuvanted vaccine, without (left side) and seven days after challenge infection with 4 × 10^5^ IFU of C.tr. E (right side). 5cVAC-specific IgA, IgM, IgG as well as IgG_1_, IgG_2a_, and IgG_2b_ were still detectable in most mice (*n* = 5–7 per group) without (**a**) as well as after challenge (**b**). In control mice (that received only c-di-AMP or buffer), no 5cVAC-specific antibodies could be detected (<1 U). The exact size of each group can be found in Appendix A.

**Figure 13 vaccines-09-00609-f013:**
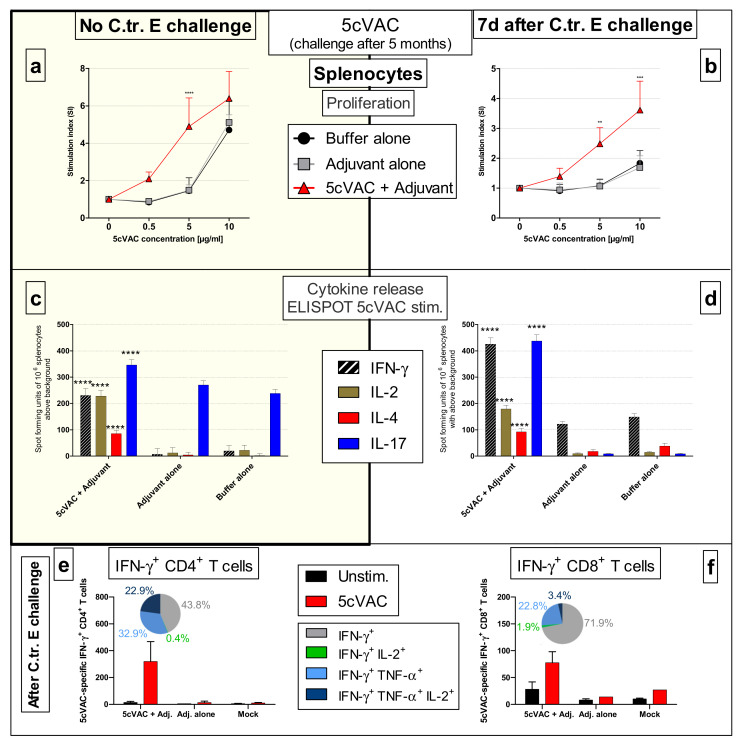
Antigen-specific cellular responses in mice, 5 months after the last i.n. booster vaccinations with adjuvanted 5cVAC. 5cVAC-induced cellular responses in the spleen were analyzed 20 weeks after the last i.n. application of the c-di-AMP-adjuvanted vaccine, without (**a**,**c**) and seven days after challenge infection with 4 × 10^5^ IFU of C.tr. E (**b**,**d**–**f**). Mice that received only c-di-AMP or buffer served as negative controls. Proliferation of splenocytes from vaccinated and control mice without (**a**) and after (**b**) challenge was determined in response to increasing concentrations of 5cVAC. The depicted stimulation indexes (SI) are the ratio of [3H]-thymidine uptake of stimulated versus non-stimulated samples. (**c**,**d**) show the dependence of the number of (ELI)Spots of IFN-γ, IL-2, IL-4, and IL-17-producing 5cVAC-restimulated cells (per 10^6^ spleen cells after subtraction of background values of unstimulated cells) on vaccination with adjuvanted 5cVAC. The results presented come from cells of *n* = 5 vaccinated and 6 non-vaccinated (buffer- or adjuvant-treated) healthy (**a**,**c**), or *n* = 7 vaccinated and 10 non-vaccinated (buffer- or adjuvant-treated) infected mice (**b**,**d**). Flow cytometric identification of multifunctional 5cVAC-restimulated CD4^+^ (**e**) and CD8^+^ (**f**) T cells was performed in *n* = 4 vaccinated and 4 non-vaccinated mice after challenge infection. The pie charts depict the proportion of single (light grey), double (green and light blue), and triple (dark blue) cytokine producers. The bar chart indicates the frequency (mean ± SD) of single (IFN-γ), double (IFN-γ^+^/TNF-α^+^, IFN-γ^+^/IL-2^+^) and triple (IFN-γ^+^/TNF-α^+^/IL-2^+^) Th1 cytokine-producing multifunctional CD4^+^ and CD8^+^ T cells. The number of unstimulated cells was subtracted from the respective number of stimulated samples. The performed statistical analysis is described in Section 2. For statistical analysis of the cellular responses, differences from the 5cVAC restimulated versus non-stimulated splenocytes are shown by **,***,****: ** *p* < 0.01, *** *p* < 0.001 or **** *p* < 0.0001). The exact size of each group can be found in Appendix A.

**Table 1 vaccines-09-00609-t001:** Identities (%) among species and serovars of the complete proteins (Cp) and protein fragments (Pf) which were used as antigens in the vaccine. Depicted are also the protein fragments of *Chlamydia trachomatis* (C.tr.) serovar E/DK20 used for vaccination (in amino acids, aa). Identities are given in relation to those (versus, vs) of C.tr. D/UW3/CX, A/HAR-13, L2 LGV II 434, and *C. muridarum* (C.mu.). The strain names of the different serovars from which this information is derived are included in the column titles of Table 2.

	C.tr. E/DK20 Pf Antigen	C.tr. E Cp vs. C.tr. D Cp	C.tr. E Pf vs. C.tr. D Pf	C.tr. E Cp vs. C.tr. A Cp	C.tr. E Pf vs. C.tr. A Pf	C.tr. E Cp vs. C.tr. L2 Cp	C.tr. E Pf vs. C.tr. L2 Pf	C.tr. E Cp vs. C.mu. Cp	C.tr. E Pf vs. C.mu. Pf
PmpA	53aa-665aa	99.8%	99.7%	99.8%	99.8%	100.0%	100.0%	76.8%	81.2%
PmpD	34aa-619aa	99.1%	99.5%	99.0%	99.5%	99.4%	99.7%	71.7%	68.6%
PmpG	29aa-673aa	99.4%	99.5%	99.9%	99.7%	97.5%	97.7%	72.8%	67.2%
PmpH	26aa-690aa	99.7%	99.6%	94.7%	92.5%	94.7%	92.9%	75.4%	69.1%
Ctad1	1aa-433aa	100.0%	100.0%	98.6%	98.6%	98.4%	98.4%	82.0%	82.0%

## Data Availability

The raw data supporting the conclusions of this article will be made available by the authors, without undue reservation.

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
