# Peer review of "Prophylactic Multi-Subunit Vaccine against Chlamydia trachomatis: In Vivo Evaluation in Mice"

_vaccines, 2021, doi:10.3390/vaccines9060609_

Round 1

Reviewer 1 Report

In this manuscript the authors explore the use of a multivalent vaccine approach for targeting C. trachomatis in a murine infection challenge model. There use of polymorphic membrane proteins as targets is innovative, as many are well conserved across serovars of C. trachomatis, and have the potential to provide protection against a number of strain types, compared to vaccines that utilize MOMP antigens, which are often serotype specific. The authors also investigate the use of c-di-AMP for use as an adjuvant in their vaccine approach.

The study is interesting, but there are a number of concerns regarding the experimental design. For example, it is unclear why the authors chose to use a respiratory infection route for these studies. All of the strains examined are either genital or ocular isolates (albeit they colonize poorly in murine, lower genital tract infection models). The respiratory model they use would be better suited for studying C. psittaci, C. pneumoniae, or even C. muridarum (which was originally isolated from murine lungs). As the authors previous work was on C. psittaci, this is likely an explanation for why they replicated their model for C. trachomatis, but it is a tough sell pitching this as a relevant infection model for an ocular / genital pathogen. 

The model also lacks a means of observing bacterial burden over time (gDNA, ifu / cfu), something that the genital infection model offers with the collection of mucosal secretions throughout an infection. Antibody tittering of secretions from the genital tract would also be more convincing that their intra nasal vaccination route is sufficient to confer protection against genital infections.

Overall, the infection model and the makes this data less interesting then it would be otherwise: it would have been encouraging if the authors had demonstrated that they could achieve mucosal immunity in the genital tract from a vaccination of the respiratory tract. There are also issues with the use of proper controls in some of the serovars examined (comparing vaccinated group with buffer instead of the adjuvant control. 

Major comments:

Line 28-29: This is not technically correct. As IgA levels in serum are low, it is not at all surprising that serum transfer did not confer protection against a mucosal pathogen.

Line 326: As this C. trachomatis model of infection is not listed via citation and is therefore new, please include this pre-test data in your results. Include details such as ifu time courses and time required to clear infections

Line 347: Again, please include this information. Knowing how the titers were achieved is important for accurate interpretation of experimental outcomes and their relevance to clinical cases.

Line 424: Please include the pilot data. This is a new model, a new system, and it needs sufficient characterization if it is to be used in the way the authors intend.

Figure 3, c1-e2: The absence of an adjuvant control in these experiments is concerning. Comparisons to buffer is not an appropriate control. Given the data presented in D1, there appears to be protection from just the adjuvant for this serovar. This could be the case for the L2 and A serovars as well. This is why it is important that the authors address what is known about the role played by c-di-AMP during infections with C. trachomatis (Barker 2012, etc.) and how this has the potential to influence their experimental outcomes.

Line 583-584: Without the adjuvant control you cannot confirm this.

Figure 4: Similar to Figure 3, you cannot associate changes in TNF-a and IFN-g to your antigens unless you compare them with the appropriate adjuvant-only control.

Figure 5, lower righthand panel: Recommend using color instead of different symbols as they are quite small and hard to distinguish.

Line 722-723: As IgA are bound to be low in serum, it is not entirely surprising that the authors did not find serum antibodies to be protective.

Line 763-765: Bacterial load appears to only differ by half a log. Long term protection appears limited to pathology as opposed to colonization.

Minor comments:

Discussion: There are a number of studies investigating the role of c-di-AMP in response to chlamydial infections. The authors should note these studies in their discussion and interpret their results with these previous findings in mind.

Reviewer 2 Report

The submitted manuscript "Prophylactic multi-subunit vaccine against Chlamydia trachomatis: in vivo evaluation in mice" is certainly of interest and relevance. With vaccination studies such as this even small amounts of information that can help direct overall ideas positively or negatively about how to overcome the challenge of disease for which we do not have current vaccines are of value. That is where this work is most relevant and worthy.

However, this manuscript should not be published until issues with the structure and presentation are corrected.

Minor Issues:

-There are some moderate English issues that need to be fixed. A few examples:

Line 111: "the majority of them is"….should be "are"

160: should read "our own data"

Oxford comma is not consistently used. 

-Table 1 is duplicated. 

-There are random different colors of text.

-There is random underlining in text.

-There us random bolding in text.

-N values are given as ranges and need to be properly defined for each figure and experiment.

Major issues: 

This manuscript, in its current form, is extremely difficult to properly and thoroughly assess. The main issues are the somewhat rambling nature of the manuscript and the figures.

The figures are inconsistent, needlessly complex, and poorly formatted. The figures need to be completely redone in a logical and consistent manner that makes it so they can be easily understood by the reader. The coloring of text, and boxes, in the figures makes them busy and hard to figure out what the importance of the coloration is supposed to be. The figure legends are also needlessly complex and drawn out. Indeed, the figure legends, at times, contain analysis and discussion material that is not pertinent to a figure legend. Figure legends are there to explain the figure so that it can stand by itself, not so that it can explain an entire manuscript.

The relevance of much of the introduction and discussion is tangential in its relevance to the data presented in this manuscript other than to recount failures which we are well aware of and reads more like a review article. While current status of the field and historical issues can be/are relevant this introduction rambles on along various tangents until at one point finally stating that something was beyond the scope of this article. How that decision was made when other tangential items were included is unclear. 

Overall, the manuscript needs a good reevaluation of the important aspects to include and the largely tangential aspects dropped for clarity and accessibility before publication

Reviewer 3 Report

This manuscript is scientifically very interesting and completely necessary for the correct development of an effective and safe vaccine against C. trachomatis. It contributes to approach scientific knowledge to an effective and safe vaccine that protects humans against a wide variety of C. trachomatis serovar. The murine model is a suitable model for testing Chlamydiaceae vaccines in experimental phases. Candidate antigens from C. trachomatis are very suitable. The adjuvant chosen also.

This is very extensive and complete study. It evaluates in detail the immune response induced by a subcellular vaccine with 5 subunits, each of these components, as well as a subcellular vaccine with just the 2 subunits that showed the best results.

The manuscript has a suitable organization, all sections are well written with a simple and concrete style. But sometimes the same information is repeated twice unnecessarily.

The material and methods give sufficient details to replicate the proposed experimental procedures and analysis.  

Although different sections are sometimes quite extensive and in most of figure or table legends include information that has been explained in the text. I understand that this is to facilitate the interpretation of the figures, but I think and recommend authors to eliminated repeated information or summarize this information.

I suggest the following corrections to improve the text:

Table 1 is repeated twice: in Lines 169-175 and in Lines 190-195. Delete one of them.

Line-301: Add the amount of c-di-AMP use in the 2cVAC

Line-308: Express in μg de amount of c-di-AMP as you did in Line 300

Line 320: Specify how many experiments

To designate the groups in the text and in figure 1 in the same manner:

Line 322: Delete Challenge model.

Line 324: specify how many animals are use per group

Line 328: specify how many animals are use in the mock group

Line 330: Delete C.tr. E Challenge model

Line 343 : change “vaccination” for “protection”

Line 395: delete “standard”

Line 395-396: Name the sections in the same way as in the text: (a) Standard vaccination, (b) Serum transfer, (c) Long term protection

Line 337: instead of “twice á 170 μg” specify that it is done days 4,5 post hormone synchronization, and indicate the volume of serum to be transferred per animal

Line 337: delete “to 15” and specify how many animals are sera transferred

Line 346: Delete “one week later,e.i”

Line 347: homogenize: they are 30 or 31 weeks old mice?? (in figure 1 it appears 30 weeks)

Line 351: change “of the bacteria,” by “ i.n. C. tr. E-challenge”

Line 416: This explanation must be better in the text not in the figure legend

In figure 1b: Delete “or” before “final blood extraction”

Figure 1: must be situated before section 2.6 Clinical scoring

Figure 1 legends include information that has been explained in the text, I think that the repeated information could be eliminated or summarized.

I consider that paragraphs 2.11, 2.12 and 2.13 are subsections of section 20.10, and therefore I propose that they be numbered appropriately:

Line 452: 2.10.1

Line 460: 2.10.2

Line 2.13: 2.10.3

Results

Figures 2-9 legends include information that have been explained in the text, try not to repeat the same information in the figure legend. Moreover, a short title should be placed in each figure legend, not an explanation of the results that are representing. In addition, the color and shape representing each group of animals in the study should be added in these legends or within the figure3, 4, 6, itself, as it is done in figure 2, 5, 7, 8, 9.

Line 500: mice were 7 weeks not 3 weeks (as you put in fig.1)

Section 3.8: I think it would be better name as “Results in the long-term protection study”, as you done in materials and methods. This section would have two subsections (3.3.1 and 3.8.2)

Section 3.9: I think is a subsection of section of 3.8. I propose to be numbered as 3.8.2

In Line 863 and line 869: Figure 9h is mentioned, but” h” it is not shown in figure 9 and “g” is in the position of the “h”. “g” must be in the down left corner of figure 9

The Discussion of the manuscript is well argued based on the results obtained, but the conclusions are not collected in a separate paragraph. On the contrary, they are interspersed in the body of the discussion. I believe that a paragraph containing all the conclusions of this work would be very valuable. Even if the discussion continues with future experiments, including different route of infection, it would be very helpful.

Round 2

Reviewer 1 Report

The authors have made a good-faith effort to address all of my concerns.

While I continue to have reservations about their preferred model system, I do believe that they now provide sufficient characterization of the model for readers to interpret and access the model's validity and relevance. The authors also note the preliminary nature of their work, and that the logical next step will be to conduct these experiments in a genital tract infection model. I look forward to reading their follow-up study.